# Adaptive Energy Alignment for Accelerating Test-Time Adaptation

**Wonjeong Choi**[1], **Do-Yeon Kim**[1], **Jungwuk Park**[1], **Jungmoon Lee**[1], **Younghyun Park**[2], **Dong-Jun Han**[3], **Jaekyun Moon**[1]

[1] Korea Advanced Institute of Science and Technology (KAIST),
[2] Agency for Defense Development (ADD), [3] Yonsei University
[1] {dnjswjd5457,dy.kim,savertm,wndanseh}@kaist.ac.kr;
[2] dnffkf369@add.re.kr; [3] djh@yonsei.ac.kr; [1] jmoon@kaist.edu

## Abstract

In response to the increasing demand for tackling out-of-domain (OOD) scenarios, test-time adaptation (TTA) has garnered significant research attention in recent years. To adapt a source pre-trained model to target samples without getting access to their labels, existing approaches have typically employed entropy minimization (EM) loss as a primary objective function. In this paper, we propose an adaptive energy alignment (AEA) solution that achieves fast online TTA. We start from the re-interpretation of the EM loss by decomposing it into two energy-based terms with conflicting roles, showing that the EM loss can potentially hinder the assertive model adaptation. Our AEA addresses this challenge by strategically reducing the energy gap between the source and target domains during TTA, aiming to effectively align the target domain with the source domains and thus to accelerate adaptation. We specifically propose two novel strategies, each contributing a necessary component for TTA: (i) aligning the energy level of each target sample with the energy zone of the source domain that the pre-trained model is already familiar with, and (ii) precisely guiding the direction of the energy alignment by matching the class-wise correlations between the source and target domains. Our approach demonstrates its effectiveness on various domain shift datasets including CIFAR10-C, CIFAR100-C, and TinyImageNet-C.

## 1 Introduction

Despite the huge success of deep neural networks (DNNs) in various fields (Mikolov et al., 2013; Krizhevsky et al., 2017), the randomness and dynamic nature of test samples encountered during inference still pose limitations to the application of DNNs. Particularly, the performance of DNNs significantly degrades in out-of-domain (OOD) scenarios (Hendrycks & Gimpel, 2017), where domain distributions of train data and test data are different. This remains a substantial challenge from the perspectives of robustness and practicality of DNNs.

As a promising direction to address the OOD problem, test-time adaptation (TTA) has been actively studied in recent years. TTA is a paradigm that involves adapting a pre-trained model (pre-trained on source domains) using unlabeled OOD samples (from the target domain) during test-time. Source domains are typically not available during TTA due to memory and privacy issues (Wang et al., 2023; Liang et al., 2023). This assumption of the source-free property has played a pivotal role in the growing interest in TTA.

The main focus of TTA is to adapt the pre-trained model on the given target samples to improve the performance at test-time. To tackle real-world scenarios where target samples arrive sequentially in a batch-wise manner, online TTA settings have become increasingly important in recent years (Wang et al., 2023). For instance, in autonomous systems (e.g., self-driving vehicles) with dynamically evolving environments (e.g., weather, lighting), the model needs to continuously make predictions on OOD samples in an online manner. In such scenarios, the key is to sequentially adapt the model on streaming target samples to accumulate knowledge of all past data during test-time, so that the model can make more reliable predictions for future target samples. However, while many TTA methods

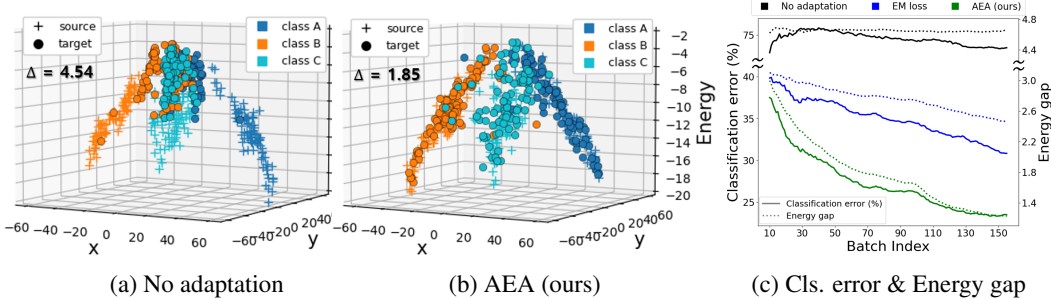

(a) No adaptation  (b) AEA (ours)  (c) Cls. error & Energy gap

Figure 1: (a-b) Sample-wise logit ($x$, $y$-axes) and energy ($z$-axis) distribution in the early stage of adaptation batches. (a) Without adaptation, target samples (CIFAR10-C) are mixed up in the high-energy region, while source samples (CIFAR10) have low energies and are clearly separated. (b) Our AEA successfully reduces the energy gap $\Delta$ by accurately guiding the direction of energy alignment. As a result, (c) our approach can accelerate test-time model adaptation and achieve remarkable performance even in a few adaptation batches.

have been proposed, they still exhibit limited performance in online adaptation settings. Particularly, they often require a large number of adaptation batches for the model to achieve a satisfactory performance, leading to suboptimal early-stage performance (e.g., on the first few batches). Our goal is to thoroughly investigate the rationale behind this phenomenon and address this issue.

**Motivation.** To overcome the aforementioned challenges, we propose a new TTA method by leveraging the idea of energy-based models (EBMs) (LeCun et al., 2006), which have proven to be effective in addressing distribution shift problems (Liu et al., 2020; Xie et al., 2022; Xiao et al., 2023; Herath et al., 2023). Their findings suggest that energy scores can serve as expressive indicators for detecting OOD samples. We start with Fig. 1a, where we visualize sample-wise logits and energy distributions of source/target samples in our TTA scenario. From Fig. 1a, we see that a source pre-trained model yields higher energies for the domain-shifted target samples and lower energies for the source samples, resulting in a notable *energy gap*. Our objective is to reduce this energy gap, aiming to alleviate domain disparity by aligning the feature representation of the target domain with that of the source domain and achieve performance improvements.

However, one key aspect that needs to be considered during the energy gap reduction is the *energy alignment direction*: In order to mitigate the domain discrepancy in a class-wise manner (in Fig. 1a), each target sample should be aligned in a distinct direction based on its class information. Addressing these energy gap and alignment issues is crucial in many TTA scenarios due to the significant domain disparity between the source and target datasets, particularly in the early stage of batch arrival. Although Yuan et al. (2024) recently introduced an energy-based TTA method, it does not consider the directional aspects of energy alignment, which limits its adaptation performance. To the best of our knowledge, these aspects have been largely overlooked in existing TTA research and remain unresolved. We thus aim to fill this gap by answering the following research question:

*Under the TTA setting where no knowledge of the source domain is available, how can we effectively align the energies between source and domain-shifted target samples while considering their class information, to accelerate performance enhancements?*

**Key idea.** In this paper, we propose adaptive energy alignment (AEA), a new energy-based TTA method that achieves fast online adaptation. We first show that the conventional entropy minimization (EM) loss can be decomposed into two energy-based terms with conflicting roles. This hinders the energy gap reduction when the EM loss is used alone, resulting in limited performance not only in the early-stage of batch arrivals but also throughout the entire online batches (in Fig. 1c and Sec. 3.2). To address this, we introduce an energy alignment scheme that strategically reduces the energy gap between the source and target domains during TTA, based on the following ideas: (i) aligning the overall energy levels (i.e., *magnitude*) of target samples with those of source domain, which the pre-trained model is already familiar with, and (ii) precisely guiding the *direction* of energy alignment by considering the structural relations between different classes, a facet that has been less configured in previous works. As depicted in Fig. 1b-1c, by leveraging well-trained knowledge of source domain in the pre-trained model, our approach robustly reduces the energy gap within a few

adaptation batches (i.e., fast adaptation) while maintaining class-wise correlations between the source and target domain.

**Summary of contributions.** Overall, our AEA is equipped with advanced optimization losses with two distinct roles for energy alignment during TTA. We summarize our contributions as follows:

- Based on the energy-based re-interpretation of EM loss, we propose a source-free energy alignment (SFEA) loss for fast TTA. The key idea is to strategically align the overall energy magnitudes between source and target domains, aiming to effectively reduce the energy gap during adaptation.

- To guide the energy alignment direction of each target sample in the logit space, we introduce a logit cosine similarity (LCS) loss as a key component. By taking advantage of a well-trained classifier's weight matrix, our LCS loss ensures that the class-wise correlation of the target domain aligns well with that of the source domain during energy alignment.

Consequently, our AEA is able to reduce the energy gap between target and source samples (SFEA loss) while considering each test sample's energy alignment direction (LCS loss). Since AEA only modifies the loss function, it introduces merely negligible extra delay/resources, as demonstrated in Sec. B.7. Our approach demonstrates remarkable performance across various domain shift datasets, including CIFAR10-C, CIFAR100-C, and TinyImageNet-C.

## 2  RELATED WORKS

**Test-time adaptation (TTA).** To tackle the TTA problem, several works (Liang et al., 2020; Wang et al., 2022) have employed pseudo-labeling techniques, where the most probable labels for unlabeled target samples are used for supervised model adaptation. Sun et al. (2020) have incorporated self-supervised learning schemes for pre-training the model. Afterwards, these auxiliary tasks are leveraged for model adaptation at test-time. Also, normalization calibration methods (Schneider et al., 2020; Wang et al., 2021; Niu et al., 2023) have been suggested to adapt normalization layers (e.g., batch/instance/group normalization) which encode distribution-specific knowledge. Other methods such as consistency regularization within augmentations (Zhang et al., 2022) and class prototype-based approaches (Iwasawa & Matsuo, 2021) have been also proposed. Regarding TTA settings, online test-time adaptation (OTTA) (Wang et al., 2023) has received attention motivated by many practical scenarios in which target samples arrive sequentially in a batch-wise manner. The goal is to continuously adapt the model to the target samples in an online fashion.

As a fundamental line of works, researchers have typically employed entropy minimization (EM) (Liang et al., 2020; Wang et al., 2021; Niu et al., 2022; Zhang et al., 2022; Niu et al., 2023) as a primary objective function to adapt the model using unlabeled target samples. While the EM loss is effective in TTA, our energy-based re-interpretation of the EM loss (in Sec. 3.2) reveals its potential hindrance to the model's assertive adaptation. Specifically in OTTA scenarios, existing works exhibit sub-optimal early-stage performance, resulting from the insufficient adaptation to the target domain. In contrast, our AEA improves both the early-stage and overall performance in OTTA by explicitly reducing the energy gap between the source and target domains.

**Energy-based models (EBMs).** EBMs (LeCun et al., 2006) adopted a score-based learning approach, allowing them to model various types of distributions in a flexible way. Such effectiveness in modeling distribution has enabled the utilization of EBMs in complex tasks such as image generation (Du & Mordatch, 2019; Du et al., 2020) and generative classifiers (Larochelle & Bengio, 2008; Yang & Ji, 2021) in a high-dimensional space. Also, some researchers have realized that modern discriminate models can be interpreted as EBMs as well (LeCun et al., 2006; Grathwohl et al., 2020; Liu et al., 2020). EBMs have been also utilized to address various distribution shift problems such as out-of-distribution detection (Liu et al., 2020), anomaly detection (Du et al., 2022), domain adaptation (Zou et al., 2021; Xie et al., 2022; Herath et al., 2023) and domain generalization (Du et al., 2022; Xiao et al., 2023). Recently, Yuan et al. (2024) have suggested an energy-based TTA approach that decreases the target energies within the model's distribution to improve model generalizability. However, it focuses solely on reducing the overall energy level without considering the energy alignment direction and requires multiple iterations to generate negative samples during adaptation. This potentially limits its performance and practicality, as demonstrated in Sec. 4 and Sec. B.7.

To address the shortcomings of existing works, we propose a new AEA scheme that not only aligns energy levels (i.e., *magnitudes*) between source/target domains but also considers *energy alignment direction* to further improve adaptation performance. Drawing insight from revisiting the EM loss, our scheme introduces two key loss functions that accelerate TTA in a challenging source-free TTA setup where the source domains are not available at test-time. Our approach introduces negligible computational costs during adaptation, enhancing its practicality in a wide range of applications.

## 3 AEA: ENERGY-LEVEL ALIGNMENT FOR TTA

### 3.1 PROBLEM SETUP AND PRELIMINARY

**Settings.** Let $D_S = \{\mathcal{X}_S, \mathcal{Y}_S\}$ be the source data sampled from the source domain distribution $P^S(x, y)$, and $D_T = \{\mathcal{X}_T, \mathcal{Y}_T\}$ be the target data from the target domain distribution $P^T(x, y)$, where $P^S(x, y) \neq P^T(x, y)$ but they have common label sets. Each pair $(x_i, y_i) \in \mathcal{X} \times \mathcal{Y}$ of samples $x_i$ and labels $y_i$ from the source/target domains follow the joint distribution $P^S(x, y)$ /$P^T(x, y)$, respectively. Given a discriminate model $f^S$ pre-trained on the source data $D_S$, TTA aims to adapt the model to the target domain $P^T(x, y)$. In our online settings, it is required to perform adaptation (i) without access to source data and (ii) using unlabeled target samples (or batches), which are incoming in a sequential manner. Specifically, for current test time step $i$, the model $f^S$ adapts to mini-batch $B_i = \{x_j\}_{j=1}^{|B|}$, consisting of domain-shifted, unlabeled target samples $x_j \sim P^T(x)$ with the size of $|B|$. To make predictions for the next batch $B_{(i+1)}$, the model needs to persistently adapt in an online manner by accumulating knowledge from past batches (i.e., $B_1, B_2, ..., B_i$).

**Energy-based models (EBMs).** By defining an energy function $E_\theta(x) : \mathbb{R}^D \mapsto \mathbb{R}$ that maps input $x \in \mathbb{R}^D$ to non-probabilistic scalar called the *energy*, one can design energy-based models (EBMs) (LeCun et al., 2006) with neural networks parameterized by $\theta$. Following the energy-based interpretation of discriminate models (Grathwohl et al., 2020; Liu et al., 2020), we consider a neural network $f_\theta(x)$ modeling a categorical distribution with a softmax function $p_\theta(y|x) = \frac{\exp(f_\theta(x)[y])}{\sum_{y'} \exp(f_\theta(x)[y'])}$, where $f_\theta(x)[y']$ denotes the network output (i.e., logit) for input $x$ and class $y'$. Also, we can define the energy function $E_\theta(x)$ (also known as the *free energy*) with a log partition function as

$$E_\theta(x) = -\log \sum_y \exp(f_\theta(x)[y]). \tag{1}$$

Detailed derivations can be found in Sec. A.1. The energy score $E_\theta(x)$ is well-known to serve as a representative indicator for distinguishing between in-distribution (ID) and out-of-distribution (OOD) samples (Liu et al., 2020; Xie et al., 2022; Du et al., 2022; Herath et al., 2023). Since most modern discriminate models have employed negative log-likelihood (NLL) as a supervised loss, this turns out to push down the energy for ID samples, while those for OOD samples remain relatively high.

### 3.2 OBSERVATIONS AND MOTIVATIONS

As shown in Fig. 1a, an energy disparity between the source and the target domains is also observed in TTA. Even in online TTA where the model is sequentially adapted, this occurrence still becomes pronounced in the early-stage of batch arrival since the model is less adapted to the target domain with a few batches. Recently, several works (Xie et al., 2022; Xiao et al., 2023; Herath et al., 2023) have discovered that reducing the energy gap enables rapid and effective adaptation to the target domains, serving as a key factor in mitigating domain gaps. However, in TTA, it is challenging to reduce the energy gap by using only a small number of unlabeled target samples without any source data at test time. Furthermore, we notice that the conventional loss function, entropy minimization (EM), has limitation in sufficiently aligning the energy, resulting in unsatisfactory performance. We further discuss this issue in the following section.

**Revisiting entropy minimization.** We provide an energy-based re-interpretation of entropy minimization (EM) loss to substantiate the validity of our core idea, the energy alignment. The EM loss, denoted by $\mathcal{L}_{EM}(x; \theta)$, has been widely utilized in TTA works (Liang et al., 2020; Wang et al., 2021; Niu et al., 2022; Zhang et al., 2022; Niu et al., 2023), where the expected Shannon entropy $\mathcal{H}(x; \theta)$ for sample $x$ from a target domain $p^T$ is employed as a loss function, i.e., $\mathcal{L}_{EM}(x; \theta) := \mathcal{H}(x; \theta) = -\sum_{j=1}^K p_\theta(y_j|x) \log p_\theta(y_j|x)$. Also, this can be decomposed into two

contrastive energy-based terms from a definition of softmax function and Eqn. (1) as

$$\mathcal{H}(x;\theta) = -\sum_{j=1}^{K} p_\theta(y_j|x)\Big(f_\theta(x)[y_j] - \log \sum_{i=1}^{K} e^{f_\theta(x)[y_i]}\Big) \tag{2a}$$

$$= \sum_{j=1}^{K} p_\theta(y_j|x) E_\theta(x, y_j) - E_\theta(x), \tag{2b}$$

where $E_\theta(x, y_j) = -f_\theta(x)[y_j]$ and $K$ is the number of classes. As the last term in Eqn. (2a) does not depend on $j$, the sum of the outer probabilities is equal to 1, resulting in factoring out the energy term $E_\theta(x)$. In Eqn. (2b), the first term encourages minimizing the energy $E_\theta(x, y_j)$ with proportional to the predictive probability for the corresponding class $y_j$. This makes confident predictions even more confident by increasing the logit of each class in proportion to its confidence. In contrast, the second term, the free energy $E_\theta(x)$, aims to increase the ensemble of energies, which can be considered as a penalization term that reduces the overall scales of the logits for all classes.

**Limitation of EM loss.** Unlike the typical supervised NLL loss where the predictive probability (i.e., $p_\theta(y_j|x)$ in Eqns. (2)) is allocated to the ground truth class, the EM loss reduces the energy for each class in a dispersed manner. Also, the second term even tends to increase the overall energy. This results in insufficient reduction of the energy gap between the source and target domains, potentially hindering the assertive model adaptation in TTA. To demonstrate these effects, in Fig. 2, we compare the energy gap and accuracy of several methods. The results reveal that the EM loss, which lacks the ability to sufficiently reduce the energy gap, exhibits limited accuracy, especially in the early stage of batch arrivals. On the other hand, by using our proposed scheme (denoted as *Ours*) designed to strategically reduce the energy gap, the model

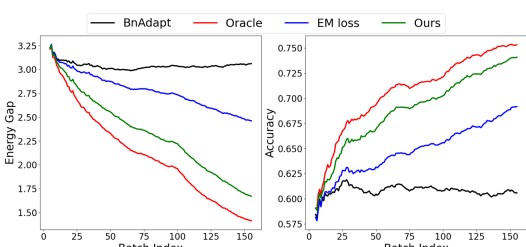

Figure 2: (left) Energy gap vs batch index; (right) Accuracy vs batch index. We carry out the online TTA on CIFAR10-C (Gaussian noise) with four different methods: (1) *BnAdapt*: Only BN parameters are updated, (2) *Oracle*: Energy alignment assuming knowledge of the source domain, (3) *EM loss*, (4) *Ours*: A combination of EM loss and our SFEA loss explained in Sec. 3.3.

adapts to the target domain more quickly, resulting in improved performance throughout all stages and getting close to the *Oracle* performance. Based on our conviction that energy alignment is a key factor in accelerating adaptation and improving performance, in the following sections, we present our AEA, a new TTA method that strategically reduces the energy gap while precisely guiding the direction of energy alignment by maintaining class-wise correlations between the source and target domains. A high-level description of our idea can be found in Fig. 3.

### 3.3 SOURCE-FREE ENERGY ALIGNMENT (SFEA) LOSS

As aforementioned, there exists an energy disparity between the source and the target domains, and conventional loss function (i.e., EM loss) has shown its limitation in reducing this gap. To accomplish this, we propose source-free energy alignment (SFEA) loss, which explicitly minimizes target energy, partially offsetting the free-energy maximization term (i.e., $E_\theta(x)$) of the EM loss in Eqn. (2b). This allows to achieve better energy gap reduction compared to using the EM loss alone. If the gap is sufficiently reduced, we can encourage the model to learn representations of target samples to become similar to those of the source samples, promoting domain-invariant representation learning.

**Formulation of SFEA loss.** To make the energy of the target samples closer to the energy of the source domain, we first estimate the energy of the source domain. One of the challenging aspects here is that we need to estimate it without access to any source information (e.g., data samples/statistics). To address this, we employ a source-like sample selection scheme, given that reliable target samples can be utilized to represent the source domain distribution to some extent (Ma et al., 2021; Du et al., 2023). Specifically, for each adaptation batch $B$, we construct a source-like batch $\hat{B}$, a set of indices of target samples with low energy scores less than the threshold $\delta_B$ defined for each batch as follows:

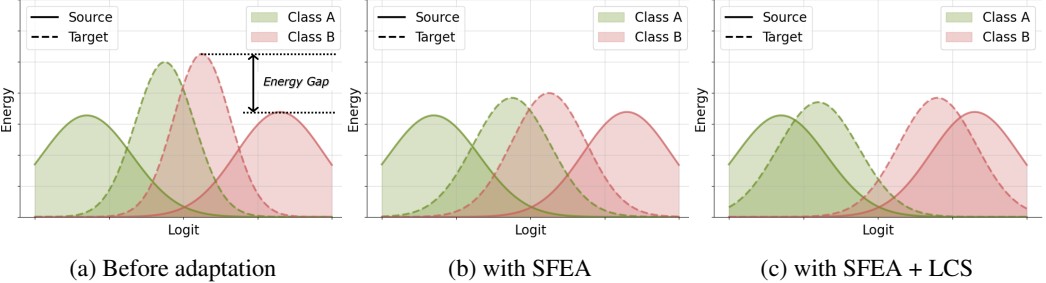

Figure 3: High-level description of our AEA. The $x$-axis represents the logit (projected), while the $y$-axis represents the energy. (a) Before adaptation, there exists the energy gap between the source and target domains, and the boundary of the two classes is fairly unclear in the target domain. (b) Our AEA sufficiently reduces the energy gap through the source-free energy alignment (SFEA) loss in Sec. 3.3. (c) Also, our logit cosine similarity (LCS) loss in Sec. 3.4 adaptively guides the direction of energy alignment in the logit space by maintaining the class-wise correlations in the target domain to those in the source domain.

$$\hat{E}_s = \frac{1}{|\hat{B}|} \sum_{i \in \hat{B}} E_\theta(x_i), \quad \hat{B} = \{i : E(x_i) \le \delta_B\}, \tag{3}$$

where $\hat{E}_s$ is the approximated energy of the source domain and $\delta_B$ is the threshold for sample selection, corresponding to the bottom $\alpha\%$ of target samples within each batch. Subsequently, we consider the averaged energy of $\hat{B}$ as the approximated energy $\hat{E}_s$ for the source domain. In light of this approximated energy $\hat{E}_s$, we introduce our source-free energy alignment (SFEA) loss $\mathcal{L}_{SFEA}$, which can penalize the target samples with higher energy $E_\theta(x)$ than $\hat{E}_s$ as follows:

$$\mathcal{L}_{SFEA}(x, \theta) = \log(1 + \exp(E_\theta(x) - \hat{E}_s)). \tag{4}$$

Notably, in Eqn. (4), we employ the softplus function (i.e., $\log(1 + e^{(\cdot)})$) (Glorot et al., 2011; Dugas et al., 2000), a differentiable and smoothed ReLU function (Agarap, 2018) as our objective. The rationale behind this choice lies in the assumption that the true energy of the source domain is likely to be slightly lower than our approximation $\hat{E}_s$, given that we approximate it using source-like target samples. Therefore, we aim to impose a small penalty even on target samples whose estimated energy $E_\theta(x)$ is less than or equal to $\hat{E}_s$. By doing so, our loss ultimately adapts the model to make the energy for each target sample closer to the approximated energy for the source domain. Although our scheme minimizes the target energy towards an adaptively changing goal (i.e., the approximated source energy) due to the infeasibility of accessing the source domain, our method turns out to align the energy of source and target domains as adaptation progresses, as shown in Fig. 1c and Fig. 2. In Sec. 4, we also demonstrate that our SFEA loss effectively reduces the energy gap and achieves attainable TTA performance.

**Key advantages of SFEA loss.** Our $\mathcal{L}_{SFEA}$ offers the flexibility to gradually adjust the degree of penalization throughout the online batch arrivals, considering the evolving domain gap between the source domain and the target samples. In other words, for the first batches where the domain gap is particularly pronounced, the energy gap (i.e., $E_\theta(x) - \hat{E}_s$) is also more significant, resulting in a stronger penalization effect. This effect steadily decreases as adaptation proceeds, until the energy gap between the source and target domains is sufficiently reduced. As a result, the model can be updated more assertively in the initial steps, accelerating adaptation and leading to significant improvement even in early-stage performance, as confirmed in Fig. 2. Moreover, compared to Yuan et al. (2024), which requires multiple iterations to decrease target energies, our SFEA loss efficiently reduces the energy gap with simple computations, requiring negligible additional costs (i.e, time delay and memory usage) as demonstrated in Sec. B.7. In the following subsection, we further take into account the directional aspects of energy alignment to achieve more robust adaptation.

### 3.4 LOGIT COSINE SIMILARITY (LCS) LOSS

Another key aspect we consider in AEA is the energy alignment direction of each target sample. This is motivated by the separated decision boundaries among classes in the target domain (see Fig. 1a and Fig. 3b). Basically, the energy function in Eqn. (1) is estimated for all possible classes, indicating

that $\mathcal{L}_{SFEA}$ reduces the energy gap in terms of overall *magnitude*. In this subsection, we propose our second key component that supplements $\mathcal{L}_{SFEA}$ by considering the energy alignment *direction*, depending on each target sample's class information. To accomplish this, we propose a logit cosine similarity (LCS) loss, which aims to accurately guide the direction of energy alignment. Our LCS can maintain class-wise correlations of target samples to those of the source domain in a logit space. We have confirmed that energy alignment in a direction that preserves class-wise correlation yields better performance, allowing target samples to be clearly distinguishable in the logit space (as in Fig. 3c).

**Key idea of LCS loss.** Specifically, our LCS loss aims to further align the logits of each target sample with those of source sample for the same class $k$, i.e., $W^{\top} z_{\text{trg}}$ is aligned towards with $W^{\top} z_{\text{src}}$, where $W$ is model's classifier weight and $z_{\text{trg}}$ and $z_{\text{src}}$ denote the output features right before the classifier for the target and source samples, respectively. This process effectively guides the direction of energy alignment in the logit space. However, the feature of source sample $z_{\text{src}}$ is not available at the time of TTA. To get around this, we bring up our idea from previous works (Liang et al., 2020; Iwasawa & Matsuo, 2021; Jang et al., 2023) that utilize the $k$-th column vector of $W$ (denoted by $W_k$) as an anchor point to make sample-wise pseudo-labels. In contrast to them, we directly employ $W_k$, which corresponds to the classifier's weight vector for class $k$, as a surrogate source feature $z_{\text{src}}$ to compose our loss function as described in Fig. 4. Since the source domain feature of class $k$ is likely to be aligned with $W_k$ to maximize the $k$-th logit during pre-training on the source domain, $W_k$ may serve as a good representative feature for $z_{\text{src}}$ of class $k$. By taking advantage of this well-trained classifier's weight matrix, our AEA is able to effectively reduce the energy gap while maintaining class-wise correlations between the source and target domains.

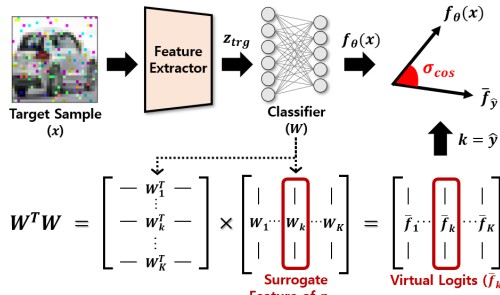

Figure 4: Description of the logit cosine similarity (LCS) loss calculated from the classifier's weight $W$.

**Formulation of LCS loss.** We define a function $\mathcal{S}(x; \theta)$ to measure the similarity of two logits as

$$\mathcal{S}(x; \theta) = \sigma_{cos}[f_\theta(x), \bar{f}_{\hat{y}}] \quad \text{where} \quad \bar{f}_{\hat{y}} = W^T W_{\hat{y}}, \tag{5}$$

where $\sigma_{cos}[\cdot, \cdot]$ denotes the cosine similarity operation. Here, $f_\theta(x)$ represents the logits for a given target sample $x$, while $\bar{f}_{\hat{y}}$ is a virtual logit computed as $W^T W_{\hat{y}}$, as illustrated in Fig. 4. The label $\hat{y}$ corresponds to the most confident class for target sample $x$. We note that the cosine similarity is used for a distance metric in that, unlike the feature representation, the weight does not inherently contain magnitude-related information; thus, the estimated virtual logit has its meaning in terms of direction rather than magnitude. Now we can formulate our LCS loss as follows:

$$\mathcal{L}_{LCS}(x; \theta) = w(x) \cdot \big(1 - S(x; \theta)\big) \cdot \mathbb{I}\{C(x; \theta) \geq C_0\}, \tag{6}$$

where $\mathbb{I}\{\cdot\}$ is an indicator function, $C(x; \theta)$ is the predictive confidence of sample $x$, and $C_0$ is pre-defined confidence threshold. In addition, $w(x)$ is a sample-wise weight, which is defined as $w(x) = \exp\big(C(x; \theta) - C_0\big)$. The weight $w(x)$ potentially encourages confident samples to have a stronger effect on LCS loss. Ultimately, the LCS loss tends to align the class-wise correlation of the target domain with that of the source domain by bringing the logit $f_\theta(x)$ of each target sample closer to the virtual logit $\bar{f}_{\hat{y}}$. Consequentially, our approach robustly guides the direction of energy alignment while maintaining class-wise correlations, thereby providing additional performance improvements.

## 3.5 OVERALL LOSS OF AEA

The overall AEA loss is defined as a combination of $\mathcal{L}_{SFEA}$, $\mathcal{L}_{LCS}$ with their respective coefficients, along with the standard unsupervised EM loss $\mathcal{L}_{EM}$ as follows:

$$\mathcal{L}_{total} = \mathcal{L}_{EM} + \lambda_1 \cdot \mathcal{L}_{SFEA} + \lambda_2 \cdot \mathcal{L}_{LCS}. \tag{7}$$

Incorporating our proposed losses ($\mathcal{L}_{SFEA}$, $\mathcal{L}_{LCS}$) consistently enhances performance and accelerates adaptation across multiple benchmark datasets (in Sec. 4). Since our method simply introduces the extra loss functions on top of the EM loss, the additional computational cost becomes negligible (in Sec. B.7). As a result, AEA can be successfully applied to a wide range of real-world scenarios, including resource-constrained applications. In the following section, we validate its effectiveness and versatility on diverse domain shift datasets.

Table 1: Classification errors (%, ↓) on CIFAR10-C with the highest severity level 5 for ResNet-26.

| Method | Noise | | | Blur | | | | Weather | | | | Digital | | | | Avg. |
|---|---|---|---|---|---|---|---|---|---|---|---|---|---|---|---|---|
| | Gauss. | Shot | Impul. | Defoc. | Glass | Motion | Zoom | Snow | Frost | Fog | Brit. | Contr. | Elastic | Pixel | JPEG | |
| No adaptation | 73.1 | 66.8 | 69.4 | 40.2 | 51.9 | 37.9 | 38.1 | 25.5 | 39.3 | 39.4 | 10.1 | 55.2 | 26.7 | 61.5 | 31.0 | 44.4 |
| TENT | 32.7 | 30.2 | 39.1 | 15.7 | 35.9 | 18.2 | 16.1 | 21.8 | 23.1 | 19.4 | 12.7 | 16.0 | 26.1 | 22.3 | 30.4 | 24.0 |
| BN adaptation | 39.1 | 36.6 | 46.1 | 17.2 | 41.1 | 19.8 | 18.0 | 25.1 | 25.5 | 21.0 | 14.1 | 17.6 | 28.9 | 26.6 | 35.5 | 27.5 |
| SHOT | 29.5 | 26.5 | 35.4 | 14.6 | 33.7 | 17.0 | 14.6 | 19.8 | 22.2 | 17.8 | 11.8 | 16.2 | 24.6 | 20.4 | 26.9 | 22.1 |
| T3A | 65.3 | 59.6 | 65.5 | 37.0 | 47.7 | 34.5 | 34.0 | 24.8 | 36.6 | 34.6 | 10.1 | 50.0 | 25.1 | 51.8 | 29.8 | 40.4 |
| TTT | 24.9 | 23.0 | 30.2 | 13.5 | 34.6 | 20.4 | 15.9 | 19.3 | 17.9 | 14.1 | 9.4 | 26.4 | 23.7 | 16.0 | 23.8 | 20.9 |
| NOTE | 48.8 | 42.5 | 47.5 | 25.0 | 40.9 | 24.4 | 23.7 | 20.3 | 23.5 | 22.8 | 9.1 | 31.6 | 24.7 | 41.9 | 29.2 | 30.4 |
| Conjugate PL | 32.7 | 30.1 | 39.2 | 15.8 | 35.8 | 18.2 | 16.1 | 21.8 | 23.1 | 19.4 | 12.7 | 16.1 | 26.1 | 22.3 | 30.4 | 24.0 |
| EATA | 38.7 | 36.2 | 46.0 | 17.1 | 40.8 | 19.8 | 17.9 | 24.9 | 25.4 | 20.9 | 14.1 | 17.6 | 28.7 | 26.4 | 35.3 | 27.3 |
| SAR | 32.9 | 30.6 | 39.5 | 15.9 | 36.3 | 18.3 | 16.5 | 22.1 | 23.2 | 19.3 | 12.7 | 16.5 | 26.2 | 22.8 | 30.6 | 24.2 |
| TEA | 27.7 | 25.5 | 34.2 | 15.3 | 34.8 | 18.0 | 15.9 | 20.0 | 20.5 | 17.7 | 12.4 | 16.4 | 25.7 | 19.2 | 26.3 | 22.0 |
| **AEA (ours)** | **24.7** | **22.7** | 31.9 | 13.8 | **30.8** | **16.2** | 13.8 | **17.5** | **17.5** | 15.4 | 10.8 | **13.7** | 23.3 | 17.4 | 23.8 | **19.5** |

Table 2: Classification errors (%, ↓) on CIFAR100-C with the highest severity level 5 for ResNet-26.

| Method | Noise | | | Blur | | | | Weather | | | | Digital | | | | Avg. |
|---|---|---|---|---|---|---|---|---|---|---|---|---|---|---|---|---|
| | Gauss. | Shot | Impul. | Defoc. | Glass | Motion | Zoom | Snow | Frost | Fog | Brit. | Contr. | Elastic | Pixel | JPEG | |
| No adaptation | 89.3 | 88.3 | 91.0 | 67.2 | 63.5 | 60.8 | 59.6 | 56.1 | 62.3 | 67.6 | 42.7 | 84.4 | 50.8 | 85.5 | 60.9 | 68.7 |
| TENT | 65.5 | 64.7 | 65.1 | 43.9 | 58.1 | 47.0 | 43.6 | 56.2 | 54.1 | 52.3 | 42.9 | 49.5 | 51.4 | 50.7 | 59.6 | 53.6 |
| BN adaptation | 70.5 | 69.9 | 68.8 | 46.6 | 60.8 | 48.8 | 45.9 | 59.0 | 56.8 | 55.1 | 45.5 | 51.2 | 53.5 | 54.8 | 62.8 | 56.7 |
| SHOT | 58.6 | 57.7 | 58.7 | 41.4 | 55.0 | 44.1 | 41.3 | 51.9 | 49.9 | 48.6 | 41.0 | 48.6 | 48.9 | 46.4 | 56.0 | 49.9 |
| T3A | 89.3 | 88.4 | 90.4 | 64.8 | 60.9 | 59.9 | 57.3 | 57.2 | 61.2 | 65.3 | 43.2 | 82.5 | 50.0 | 82.9 | 60.4 | 67.6 |
| TTT | 63.7 | 63.2 | 65.1 | 43.9 | 57.2 | 49.9 | 43.4 | 54.1 | 50.8 | 49.7 | 38.7 | 70.2 | 49.7 | 45.7 | 56.1 | 53.4 |
| NOTE | 76.4 | 74.6 | 74.5 | 53.9 | 57.6 | 50.7 | 47.9 | 52.7 | 52.3 | 56.7 | 38.6 | 67.4 | 49.0 | 70.4 | 57.8 | 58.7 |
| Conjugate PL | 65.6 | 64.7 | 65.1 | 43.9 | 58.1 | 47.0 | 43.6 | 56.2 | 54.1 | 52.3 | 42.9 | 49.5 | 51.4 | 50.7 | 59.6 | 53.6 |
| EATA | 68.0 | 66.2 | 71.5 | 46.0 | 64.7 | 49.3 | 46.0 | 56.7 | 57.2 | 53.8 | 44.1 | 51.9 | 55.4 | 52.1 | 62.2 | 56.3 |
| SAR | 65.8 | 64.9 | 65.3 | 44.2 | 58.2 | 47.1 | 43.8 | 56.4 | 54.4 | 52.5 | 43.0 | 49.3 | 51.4 | 50.8 | 59.7 | 53.8 |
| TEA | 64.0 | 63.3 | 64.2 | 45.1 | 59.0 | 48.4 | 45.4 | 56.5 | 55.1 | 52.5 | 43.3 | 53.4 | 52.7 | 50.1 | 60.0 | 54.2 |
| **AEA (ours)** | **58.2** | 58.8 | 59.0 | **40.9** | **55.0** | **43.9** | **40.5** | **51.3** | **49.0** | **47.4** | 39.4 | **44.1** | 48.4 | 44.3 | 54.8 | **49.0** |

# 4 EXPERIMENTS

## 4.1 EXPERIMENTAL SETTINGS

In this section, we demonstrate the effectiveness of our approach through extensive experiments. We follow the online TTA setting (Zhao et al., 2023) for experiments. Given a pre-trained model on the source domain, we adapt the model to the incoming batches consisting of samples from the target domain in an online manner. Specifically, the model is persistently adapted to the target domain by accumulating knowledge from past batches, to make predictions for the following batches.

**Datasets.** For the domain shift datasets, we utilize corrupted image datasets: CIFAR10-C, CIFAR100-C, and TinyImageNet-C. The uncorrupted datasets (i.e., CIFAR10, CIFAR100, TinyImageNet) are used as the source domain, while corrupted datasets (i.e., CIFAR10-C, CIFAR100-C, TinyImageNet-C) with the highest severity (i.e., level 5) serve as the target domain. We also evaluate our method on style shift dataset (i.e., PACS) in Sec. 4.2 and ImageNet to ImageNet-C dataset in Sec. B.1.

**Baselines.** To validate the effectiveness of our method, we compare AEA with the following baselines: SHOT (Liang et al., 2020), TENT (Wang et al., 2021), BN adaptation (Schneider et al., 2020), T3A (Iwasawa & Matsuo, 2021), TTT (Sun et al., 2020), NOTE (Gong et al., 2022), Conjugate PL (Goyal et al., 2022), EATA (Niu et al., 2022), SAR (Niu et al., 2023) and TEA (Yuan et al., 2024). These are chosen to cover the diverse TTA methods described in Sec. 2.

**Implementation details.** For a fair comparison, we follow the official setup of the recent TTA benchmark (Zhao et al., 2023) and set the hyperparameters for each baseline consistent with them. For CIFAR10/100 pre-training before the TTA stage, we adopt a self-supervised learning scheme (i.e., the rotation prediction) with an auxiliary head to fairly compare with an auxiliary task-based methods like TTT (Sun et al., 2020), while TinyImageNet pre-training follows the official TorchVision setup (maintainers & contributors, 2016). For performance metric, we use classification error (%, ↓) averaged over all online batches. We utilize ResNet-26 (He et al., 2016) for CIFAR10-C/100-C and ResNet-50 for TinyImageNet-C, with additional experiments under different backbones (i.e., Vision transformer (Dosovitskiy et al., 2021)) in Sec. B.2. During TTA, only batch normalization parameters are updated. Further implementation details are in Sec. C of Appendix.

Table 3: Classification errors (%, ↓) on TinyImageNet-C with the highest severity level 5 for ResNet-50.

| Method | Noise | | | Blur | | | | Weather | | | | Digital | | | | |
| | Gauss. | Shot | Impul. | Defoc. | Glass | Motion | Zoom | Snow | Frost | Fog | Brit. | Contr. | Elastic | Pixel | JPEG | Avg. |
|---|---|---|---|---|---|---|---|---|---|---|---|---|---|---|---|---|
| No adaptation | 96.6 | 95.1 | 97.2 | 92.5 | 92.2 | 77.8 | 78.5 | 81.9 | 78.1 | 89.5 | 77.8 | 98.3 | 69.5 | 71.9 | 55.6 | 83.5 |
| TENT | 66.5 | 64.3 | 72.9 | 63.2 | 75.1 | 55.5 | 54.8 | 63.6 | 61.9 | 67.6 | 57.6 | 86.2 | 56.6 | 51.5 | 52.5 | 63.3 |
| BN adaptation | 68.8 | 66.7 | 75.7 | 65.0 | 77.1 | 56.4 | 55.8 | 64.9 | 63.4 | 71.0 | 59.0 | 86.5 | 57.7 | 52.2 | 53.3 | 64.9 |
| SHOT | 64.5 | 62.8 | 70.6 | 61.9 | 73.5 | 54.5 | 53.8 | 62.1 | 61.1 | 65.2 | 56.2 | 89.6 | 55.4 | 50.8 | 51.8 | 62.2 |
| T3A | 96.6 | 95.0 | 97.3 | 92.7 | 92.1 | 77.3 | 78.2 | 82.3 | 78.2 | 89.9 | 77.1 | 98.5 | 68.9 | 70.3 | 55.8 | 83.4 |
| NOTE | 83.4 | 80.4 | 86.4 | 81.9 | 85.7 | 65.4 | 64.1 | 69.6 | 67.4 | 79.8 | 64.2 | 96.0 | 61.2 | 56.0 | 53.1 | 73.0 |
| Conjugate PL | 67.4 | 65.3 | 74.0 | 63.9 | 76.1 | 55.8 | 55.3 | 64.3 | 62.7 | 68.6 | 58.1 | 86.2 | 57.1 | 51.8 | 53.0 | 64.0 |
| EATA | 65.1 | 63.6 | 69.7 | 62.3 | 74.2 | 55.3 | 54.1 | 61.3 | 61.0 | 63.9 | 55.4 | 90.8 | 56.1 | 51.1 | 52.2 | 62.4 |
| SAR | 67.2 | 65.2 | 73.6 | 63.9 | 75.9 | 55.8 | 55.2 | 64.1 | 62.6 | 68.4 | 58.0 | 86.0 | 57.1 | 51.8 | 52.9 | 63.8 |
| TEA | 66.1 | 64.6 | 72.3 | 63.2 | 75.1 | 55.8 | 54.9 | 63.1 | 61.2 | 65.9 | 56.9 | 85.3 | 56.9 | 51.8 | 52.9 | 63.1 |
| **AEA (ours)** | **63.7** | **62.1** | **67.9** | **60.9** | **72.8** | **53.8** | **52.9** | **60.1** | **58.9** | **61.9** | **54.2** | 90.3 | **54.5** | **49.8** | **51.1** | **61.0** |

## 4.2 EXPERIMENTAL RESULTS

**Overall performance.** In Tables 1-3, we report the classification error (%) for the online TTA on CIFAR10-C, CIFAR100-C and TinyImageNet-C dataset, respectively. Experiments are conducted over 3 random trials, and we include the *full version with standard deviations* in Sec. B.9 of Appendix. They show that our AEA consistently achieves superior performances than other baselines for most of corruption types. This supports our claim that reducing the energy gap between the source and target domains contributes to the improvement of TTA performance. In several cases, TTT (Sun et al., 2020), which performs a self-supervised learning during the pre-training stage, shows competitive performances. However, this approach has limitations in a practical usage since it necessarily requires a pre-trained auxiliary head during TTA. In contrast, our AEA does not need to employ this additional head at the time of TTA. The results demonstrate that our AEA can effectively adapt models via the proposed loss functions, improving TTA performance on various domain shift datasets.

**Fast adaptation of AEA.** One of the key advantages of our AEA is its ability to accelerate model adaptation, enabling fast TTA with high performance. This can be done by explicitly reducing the energy gap through our SFEA loss. Simultaneously, our LCS loss further contributes to forming a robust decision boundary in the domain-invariant latent space to clearly separate target samples according to their classes. In Fig. 5, we report the temporal TTA performance over adaptation batches. The results show that, compared to other competitive TTA baselines, our AEA consistently

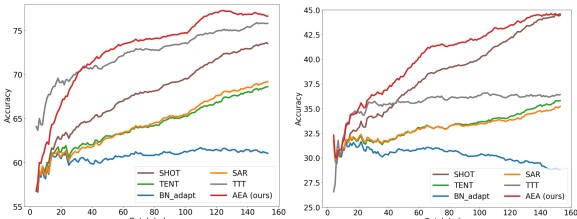

Figure 5: Fast adaptation of our AEA. We report the model accuracy (%, ↑) over adaptation batches on CIFAR10-C and CIFAR100-C datasets (with Gaussian noise). Our AEA consistently achieves superior performance, particularly in the early stages of batches.

achieves superior performance, especially in the early stages of test batches from the target domain. Furthermore, the significant performance gaps between our AEA and TENT (i.e., only EM loss) demonstrate the effectiveness of our proposed objective functions in accelerating TTA.

**Result for style shift dataset.** In Table 4, we present the results on the style shift dataset, PACS (Li et al., 2017) using ResNet-50. Each column represents the average classification error for the corresponding target domain, with the remaining domains used as independent source domains (i.e, single-domain generalization setting). Our AEA consistently outperforms other baselines, demonstrating its effectiveness across diverse types of domain shifts. This highlights our AEA's versatility in various domain shift scenarios.

| Methods | | P | A | C | S | Avg. |
|---|---|---|---|---|---|---|
| No adapt. | | 24.67 | 42.74 | 48.26 | 42.18 | 39.46 |
| TENT | | 16.55 | 26.23 | 29.65 | 34.25 | 26.67 |
| BN adapt. | | 16.72 | 26.68 | 30.18 | 36.57 | 27.54 |
| T3A | | 16.46 | 37.98 | 33.88 | 36.05 | 31.09 |
| SAR | | 16.63 | 26.41 | 29.87 | 34.98 | 26.97 |
| **AEA (ours)** | | **16.22** | **24.32** | **27.22** | **30.76** | **24.63** |

Table 4: Results on style shift dataset (PACS) with 4 different styles: Photo, Art painting, Cartoon and Sketch.

### 4.3 Further Studies on AEA

**Effect of each component.** In Table 5, we conduct an ablation study on each component of our AEA, demonstrating that both proposed loss functions (i.e., $\mathcal{L}_{SFEA}$, $\mathcal{L}_{LCS}$) independently contribute to performance improvement. Compared to using only the EM loss, the SFEA loss significantly enhances TTA performance by accelerating adaptation, while the LCS loss further makes the direction of energy alignment robust, leading to additional gains.

| Methods | CIFAR10-C | CIFAR100-C |
|---|---|---|
| No adaptation | 44.4 | 68.7 |
| EM | 24.0 | 53.6 |
| + SFEA | 20.7 | 51.8 |
| **+ SFEA + LCS** | **19.5** | **49.0** |

Table 5: Effect of each component in our AEA.

**Effect of hyperparameters.** In Fig. 6, we perform an ablation on the hyperparameters introduced in AEA, including $\lambda_1$, $\lambda_2$, $\alpha$, and $C_0$, to evaluate the robustness of our method to varying hyperparameter settings. Each subfigure represents our AEA's performance across a specified range of a given hyperparameter, while keeping the others fixed at their default values. As shown in the results, AEA consistently outper-

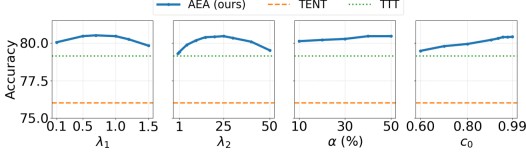

Figure 6: Ablation on varying hyperparameters for our AEA. We compare our method with competitive baselines through the average accuracy on CIFAR10-C.

forms other baselines by a notable margin across different hyperparameter settings. This not only confirms the robustness of our method on hyperparameters but also highlights its overall superiority.

**Why does energy alignment lead to better adaptation?** We find that our energy alignment help mitigate domain disparity by promoting *feature alignment* between the source and target domains, a well-established factor for improving domain adaptation (Yang et al., 2022; Lu et al., 2022). Specifically, in our implemen-

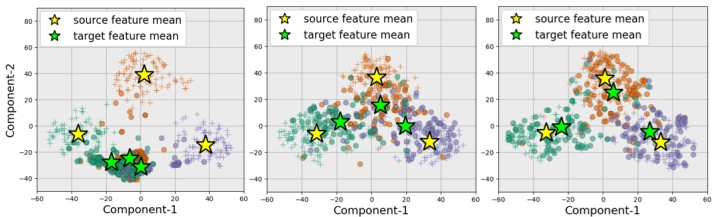

Figure 7: TSNE visualization of features for three methods: no adaptation (left), EM loss (center) and AEA (right).

tation, we freeze the classifier of the source pre-trained model while updating the batch normalization (BN) parameters of the feature extractor. As illustrated in Liang et al. (2020), the classifier retains fine-grained knowledge of the source domain, and aligning the target domain's representation with the source hypothesis facilitates OOD predictions. Similarly, our energy alignment approach encourages a reduction in target energy, which updates the model parameters to ensure that target features align with the classifier's weights (i.e., source hypothesis). This, in turn, leads to the feature alignment between the source and target domains, resulting in improved TTA performance with accelerated adaptation. To validate this, in Fig. 7, we provide feature visualizations for source and target samples in the early stage of TTA adaptation. Each color represents a different class, while the green and yellow stars indicate class-wise feature means for the target and source domains, respectively. As shown, our AEA (right) effectively aligns source and target features, leading to improved performance, which is not the case in situations without adaptation (left) and with only EM loss (center). This improvement occurs due to the fact that our energy alignment encourages the model to reduce the energies of target samples, matching them to source samples, which inherently aligns their feature representations as well. A more in-depth discussion can be found in Sec. A.8 of Appendix.

## 5 Conclusion

In this work, we propose an adaptive energy alignment (AEA) scheme to handle OOD samples in TTA setups. Based on our insight that aligning energy levels between the source and target domains facilitates rapid and effective adaptation to the target domains, our AEA strategically reduces the energy gap to enhance TTA performance. By revisiting the EM loss, we discover its limitations in reducing the energy gap. To overcome this, we introduce 1) the SFEA loss, which effectively minimizes the energy gap without access to the source domain, and 2) the LCS loss, which guides the direction of energy alignment by considering class-wise correlations. With its ability to accelerate model adaptation, our approach is versatile and leads to improved performance in various scenarios.

## REPRODUCIBILITY STATEMENT

To ensure reproducibility, we provide our implementation details and setups in Sec. 4 and Sec. C. This includes information on implementation benchmark, hyperparameters, and computing resources. Our source code is publicly available at https://github.com/wonjeongchoi/AEA.

## ACKNOWLEDGEMENTS

This work was supported in part by Samsung Electronics Co., Ltd. and by the NAVER-Intel Co-Lab. Additionally, it was funded by the National Research Foundation of Korea (NRF) through grants from the Korea government (MSIT) (No. RS-2024-00408003, RS-2024-00340966) and by the Institute of Information & communications Technology Planning & Evaluation (IITP) under MSIT of Korea (No. RS-2024-00444862). Dong-Jun Han is the corresponding author.

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

## APPENDIX

## A ADDITIONAL ANALYSIS

### A.1 DETAILED DERIVATION OF ENERGY FUNCTION

In this section, we provide detailed derivation of energy function in Sec. 3.1. By defining an energy function $E_\theta(x) : \mathbb{R}^D \mapsto \mathbb{R}$, one can design energy-based models (EBMs) (LeCun et al., 2006) with neural networks parameterized by $\theta$. From the defined $E_\theta(x)$ and the Gibbs distribution, the probability density function $p_\theta(x)$ can be derived as

$$p_\theta(x) = \frac{\exp(-E_\theta(x))}{Z(\theta)}, \qquad \forall\, x \in \mathbb{R}^D, \tag{8}$$

where $Z(\theta) = \int_x \exp(-E_\theta(x))dx$ denotes the partition function. Following the energy-based interpretation of discriminate models (LeCun et al., 2006; Grathwohl et al., 2020; Liu et al., 2020), we consider a neural network $f_\theta(x)$ modeling a categorical distribution with a softmax function $p_\theta(y|x) = \frac{\exp(f_\theta(x)[y])}{\sum_{y'} \exp(f_\theta(x)[y'])}$, where $f_\theta(x)[y']$ denotes the network output (i.e., logit) for input $x$ and class $y'$. Without changing the model $f_\theta$, we can represent the joint distribution $p_\theta(x, y)$ and density $p_\theta(x)$ marginalized over $y$ as

$$p_\theta(x, y) = \frac{\exp(f_\theta(x)[y])}{Z(\theta)}, \quad p_\theta(x) = \frac{\sum_y \exp(f_\theta(x)[y])}{Z(\theta)}, \tag{9}$$

where $Z(\theta)$ is the partition function and $E_\theta(x, y) = -f_\theta(x)[y]$. By combining Eqns. (8)-(9), we can define the energy function $E_\theta(x)$ (also known as the *free energy*) with a log partition function as

$$E_\theta(x) = -\log \sum_y \exp(f_\theta(x)[y]) \tag{10}$$

### A.2 WHY DOES EM LOSS APPEAR IN OUR FINAL LOSS?

In our AEA, we define our final loss as a combination of $\mathcal{L}_{SFEA}$, $\mathcal{L}_{LCS}$ along with the EM loss $\mathcal{L}_{EM}$ (see Eqn. (7)). Basically, the foundation of our AEA is the inherent limitations of EM loss in sufficiently reducing the energy gap in the early-stage of batch arrivals where the energy gap between source and target domains is pronounced. To address this, our SFEA loss is proposed to adaptively minimize target energy over time (strongly in the initial batches and more weakly later on), thereby reducing the energy gap more quickly, while our LCS loss considers the directional aspect of energy alignment. In turn, our method is designed to complement the limitations of the EM loss while leveraging its advantages in TTA (rather than replacing it), which is why we incorporate EM loss in our final objective.

### A.3 ENERGY-LEVEL DISCREPANCY BETWEEN THE SOURCE AND TARGET DOMAINS

In Fig. 8, we demonstrate the energy level discrepancy between the source and target domains to further strengthen our implications. As shown, the energy levels of the target samples, which is referred to as 'CIFAR10-C', are higher than that of the source samples of CIFAR10 regardless of the classes. Due to this energy level discrepancy between the source and target datasets, the existing EM loss approaches exhibit a limited performance in the early-stage of online batches as the EM loss fails to sufficiently decrease the energy levels of the target samples. In contrast, our method strategically reduces this energy gap while precisely guiding the direction of energy alignment by using our proposed losses $\mathcal{L}_{SFEA}$, and $\mathcal{L}_{LCS}$.

### A.4 MOTIVATION OF OUR LCS LOSS

Our LCS loss is proposed to address the directional aspect of energy alignment, particularly concerning the class-wise adjustment of logit elements. Specifically, our SFEA loss alone minimizes the free energy of the target samples, aligning the energy between the source and target domains in

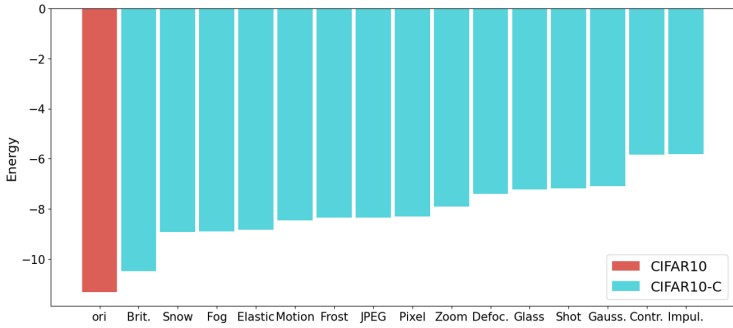

Figure 8: Energy-level discrepancy on CIFAR10 (red) and CIFAR10-C (blue) dataset at the test time.

terms of the magnitude of energy levels. It tends to increase the overall magnitude of the logits across all classes rather than considering the relativity of each logit element (i.e., class-wise correlation). On the other hand, our LCS loss additionally accounts for class-wise correlation by increasing the logit elements of classes with high correlation together. This allows each target sample to align with the well-trained source knowledge in a class-wise manner, leading to directionally improved energy alignment.

To support our claim, in Fig. 9, we provide sample-wise visualizations (similar to Figs. 1a-1b) with and without our LCS loss. In Fig. 9b, while our SFEA loss effectively reduces the energy gap, a number of target samples are still not clearly separated according to their true classes. In contrast, by incorporating our LCS loss in Fig. 9c, the target samples align more successfully with the source samples in a class-wise manner, guiding the direction of energy alignment in a logit space.

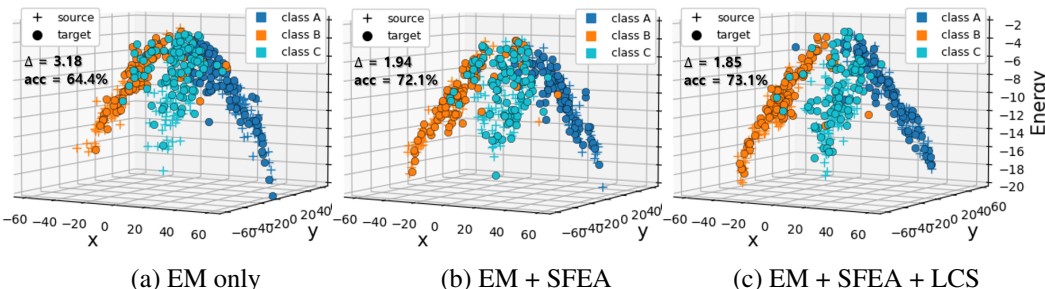

(a) EM only        (b) EM + SFEA        (c) EM + SFEA + LCS

Figure 9: Sample-wise logit ($x$, $y$-axes) and energy ($z$-axis) distribution in the early stage of adaptation steps (similar to Figs. 1a-1b). Although our SFEA loss effectively reduces the energy gap, a number of target samples are still not clearly separated according to their true classes. In contrast, by incorporating our LCS loss, the target samples align more successfully with the source samples in a class-wise manner, guiding the direction of energy alignment in a logit space.

## A.5   FURTHER ANALYSIS ON THE MODEL OUTPUT

In table 6, we present additional results that help understand how our proposed losses affect model outputs. Specifically, on CIFAR10-C dataset, we consider four methods: No Adaptation, EM, EM + SEFA, and EM + SEFA + LCS, measuring their average energy scores for source/target domains ($E_{src}$, $E_{trg}$), accuracy and entropy of test samples. From the results, we can further recognize the effects of our method as follows: Our proposed methods (i.e., $L_{SEFA}$, $L_{LCS}$) can effectively reduce the energy gap between the source and target domains, leading to improved accuracy. Furthermore, it is observed that our proposed methods produce lower entropy than TENT (EM loss) by guiding target samples to be placed in more confident and correct regions.

## A.6   ABLATION STUDY ON THE CHOICE OF LOSS FUNCTION

In Table 7, we conduct an ablation study on the choice of loss function for energy alignment, comparing the hinge loss to our SFEA loss. Specifically, we report the results with varying $\lambda_1$, a coefficient of $\mathcal{L}_{SFEA}$, to evaluate hyperparameter sensitivity. The results demonstrate that our loss is

|  | No adaptation | EM | EM+SEFA | EM+SFEA+LCS |
|---|---|---|---|---|
| $E_{src}$ | -10.26 | -10.54 | -11.26 | -10.83 |
| $E_{trg}$ | -6.85 | -7.35 | -8.81 | -9.28 |
| $|E_{src}\text{-}E_{trg}|$ | 3.41 | 3.19 | **2.45** | **1.55** |
| Accuracy (%) | 61.5 | 64.8 | **71.0** | **72.1** |
| Entropy | 7.94 | 7.84 | **7.67** | **7.65** |

Table 6: Analysis on the model output.

less sensitive to hyperparameters than hinge loss, achieving better performance with marginal gains. Our intuition is that these improvements are attributed to the smoothness of the softplus function around the near-zero region, containing more information useful for reducing the energy gap than the hinge loss near that region.

Also, unlike previous energy alignment works that have access to the source domain, the source-free scenario in TTA does not allow for accurate estimation of source energy. Under this new setting, we have utilized the approximated source energy for energy alignment, wherein our smoothed loss design proves to be more feasible and leads to better performance compared to the hinge loss.

| $\lambda_1$ | 1.0 | 1.2 | 1.4 | 1.6 | 1.8 | 2.0 |
|---|---|---|---|---|---|---|
| hinge | 80.3 | 80.2 | 79.9 | 79.7 | 79.3 | 79.0 |
| ours | **80.5** (+0.2%) | **80.7** (+0.5%) | **80.6** (+0.7%) | **80.2** (+0.5%) | **79.9** (+0.6%) | **79.8** (+0.8%) |

Table 7: Ablation study on the choice of loss function for energy alignment, comparing the hinge loss to our SFEA loss. We report the average classification accuracy (%, ↑) on CIFAR10-C dataset. The other experimental details (e.g., pre-trained model, hyperparameters, etc.) are consistent with the main settings in the paper.

### A.7 EVIDENCE FOR EM LOSS'S LIMITATION ON ENERGY GAP REDUCTION

Our argument on this topic initiates from an energy-based reinterpretation in Sec. 3.2, where the EM loss can be decomposed into two contrastive energy-based terms. Regarding the $1^{st}$ term (i.e., $\sum_{j=1}^{K} p_\theta(y_j|x)E_\theta(x, y_j)$), given the absence of labels for target samples, this term dispersely reduces the energy given class (i.e., $E_\theta(x, y_j)$) across each class, whereas the supervised NLL loss concentrates predictive probabilities on the ground truth class. Also, the $2^{nd}$ term (i.e., negative free energy, $-E_\theta(x)$) encourages the increase of ensemble of energies as a penalization during adaptation.

In TTA, due to the typically limited adaptation batches provided for sequential target batches, the EM loss exhibits limitations in sufficiently reducing the energy gap within the constrained iterations, potentially hindering the assertive adaptation in TTA. Confirmingly, as shown in Fig. 1c and Fig. 2, the EM loss alone is not sufficient to reduce the energy gap between the source and target domains. For a more comprehensive understanding, in Fig. 10, we visualize how these two decomposed terms change during TTA. Here, $-E_{src}$ and $-E_{trg}$ represent the negative free energy (i.e., $2^{nd}$ term) for the source and target domain, respectively, and the $1^{st}$ term corresponds to $\sum_{j=1}^{K} p_\theta(y_j|x)E_\theta(x, y_j)$ as denoted in Eqn. (2). In the result, among the two terms, the $1^{st}$ term is dominant, leading to a reduction in energy as adaptation progresses. However, in the case of EM loss, the energy gap between source and target domains slowly decreases, and it fails to reach the energy level of the source domain. On the other hand, by adding our SFEA loss to EM loss, we can explicitly align the energy, thereby reducing the energy gap more fast. Also, this result is consistent with Fig. 2 in our main paper; our SFEA loss facilitates a close approximation to the Oracle case, which assumes knowledge of source domain, achieving superior TTA performance.

### A.8 THE LINK BETWEEN ENERGY GAP REDUCTION AND BETTER ADAPTATION

In this paper, we argue that the performance of TTA can be improved by aligning the energy levels between the source and target domains. To accomplish this, our AEA adopts the SFEA loss (in Sec.

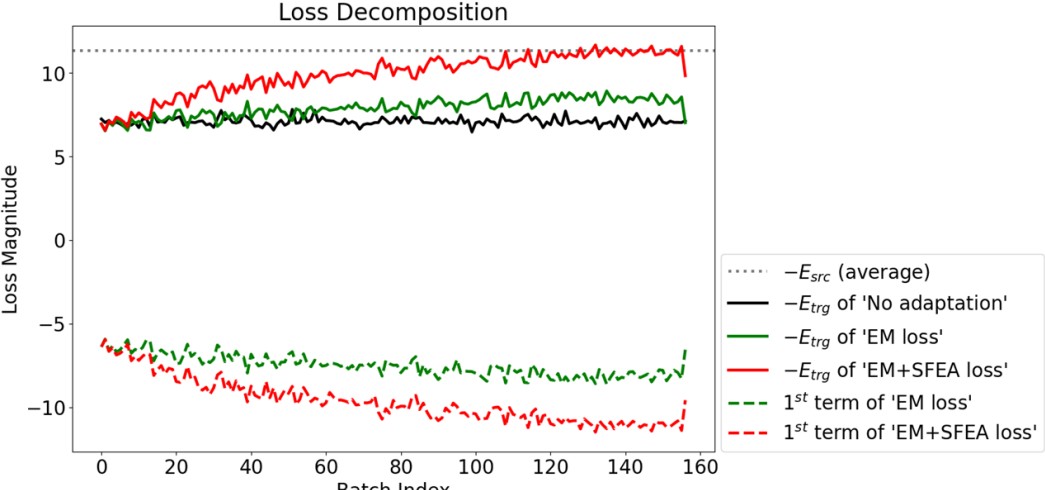

Figure 10: Visualization of how the two decomposed loss terms in Eqn. (2) change during TTA on CIFAR10-C.

3.3) and the LCS loss (in Sec. 3.4), which successfully reduce energy gap, thereby accelerating model adaptation and achieving superior TTA performance on various domain-shift datasets.

As introduced in Sec. 4.3, one can pose a following question: *why would reducing this energy gap improve TTA?* We figure out that our energy alignment can also facilitate feature alignment between the source and target domains. Indeed, it has been widely studied that the feature alignment through domain-invariant representation learning effectively addresses domain shift problems (Yang et al., 2022; Lu et al., 2022). However, as they are typically allowed to access both source and target samples, it is straightforward to align features of both domains, while the absence of source domain in our TTA setting makes it quite challenging. Instead, we utilize the energy level that obviously differentiate between the source and target domains as a surrogate feature to be aligned. We note that before adaptation, the pre-trained model produces low energy (i.e., high logits) for the source domain and high energy (i.e., low logits) for the target domain (as observed in Fig. 1a). As an alternative to directly aligning features, our AEA strategically matches the energy levels between the source and target domains in the logit space, encouraging the model to learn aligned feature representations.

More specifically, in the implementation, we freeze the classifier of pre-trained model which is trained on the source domain, while updating model parameters (e.g., BN parameters of feature extractor) prior to the classifier. As discussed in Liang et al. (2020), such a classifier holds fine-grained knowledge on the source domain (i.e., source hypothesis), and by aligning the target domain's representation with the source hypothesis, a well-trained classifier can be effectively utilized for performance improvement. In this context, since our energy alignment encourages decreasing the energies (or increasing the overall logits) of target samples, it necessarily makes the model parameters to be updated so that the target features are aligned with the classifier's weight itself, which is already sufficiently aligned with the source domain. Consequentially, this inherently results in alignment of feature representations of target domain with those of source domain, leading to performance enhancement.

In Fig. 7, we provide the feature visualization of source/target samples within a few adaptation batches (after 30 steps out of total 157 steps). As can be seen, in the cases without adaptation (left) and with just EM loss (center), the class-wise feature means are not aligned well between source and target; at the same time, these two cases have shown high energy gaps and low performances on the target domain (as shown in Fig. 1c and 2). On the other hand, our AEA (right) can significantly align features between the source and target domains, encouraged by our energy alignment approach. From the results, we emphasize that our strategical and effective energy alignment scheme through SFEA loss and LCS loss facilitates feature alignment, which leads to improved TTA performance.

For further analysis, in Table 8, we measured the true distance between source features and target features during TTA adaptation. Specifically, for each adaptation stage, we (1) estimate the source/target

| | 0 | 10 | 20 | 30 | 40 | 50 | 60 | 70 | 80 | 90 | 100 | 110 | 120 | 130 | 140 | 150 |
|---|---|---|---|---|---|---|---|---|---|---|---|---|---|---|---|---|
| No adaptation | 5.18 | 5.18 | 5.18 | 5.18 | 5.18 | 5.18 | 5.18 | 5.18 | 5.18 | 5.18 | 5.18 | 5.18 | 5.18 | 5.18 | 5.18 | 5.18 |
| TENT | 5.18 | 1.87 | 1.83 | 1.80 | 1.76 | 1.73 | 1.69 | 1.64 | 1.61 | 1.58 | 1.55 | 1.52 | 1.49 | 1.46 | 1.44 | 1.41 |
| AEA (ours) | 5.18 | **1.58** | **1.28** | **1.09** | **0.98** | **0.94** | **0.89** | **0.86** | **0.82** | **0.79** | **0.78** | **0.77** | **0.75** | **0.71** | **0.69** | **0.71** |

Table 8: Feature alignment between the source and target domains. We measure the true distance (i.e, Euclidean distance) between source features and target features during TTA adaptation. Each column represents the number of adaptation batches. Experiments are conducted on CIFAR10-C (gaussian noise with severity level 5).

feature means for each ground-truth class, (2) measure the Euclidean distance between the feature means for each class, and (3) take the average across all classes. These measurements can indicate how well the adapted model performs feature alignment between the source and target domains. The results demonstrate that our method indeed aligns the true source and target features more rapidly. For instance, our AEA achieves better alignment by reducing the distance between each feature from 5.18 to 0.98 over 40 adaptation batches, whereas TENT does from 5.18 to 1.76. Based on these findings, we argue that our adaptive energy alignment approach facilitates more effective feature alignment from the early online batches, accelerating TTA adaptation.

# B ADDITIONAL RESULTS

## B.1 EVALUATION ON IMAGENET-C DATASET

In Tables 9-10, we evaluate the online TTA performance on ImageNet-C dataset, comparing our AEA with comparable baselines. For the experiments, we employ ResNet-50 backbone as a pre-trained model and follow the setup of TENT (Wang et al., 2021). We consider two different practical settings using the large-scale ImageNet-C dataset, which contains 50K target samples: First, we assume there are sufficient target samples (50K, full), and second, we examine a case where the number of target samples is smaller (10K). We report each result in Table 9 and Table 10, respectively. The results show that our AEA still achieves competitive performance in both cases. Notably in case of limited target samples (10K), our AEA outperforms the baselines by large margins ($\geq 4.0$). This result indeed supports our claim that our adaptive energy alignment approach can achieve desired performance even in a few adaptation batches compared to the baselines, enhancing its applicability in various scenarios.

Table 9: Classification errors (%, ↓) on ImageNet-C (50K target samples) with the highest severity level 5 for ResNet-50 backbone.

| | Noise | | | Blur | | | | Weather | | | | Digital | | | | |
|---|---|---|---|---|---|---|---|---|---|---|---|---|---|---|---|---|
| Method | Gauss. | Shot | Impul. | Defoc. | Glass | Motion | Zoom | Snow | Frost | Fog | Brit. | Contr. | Elastic | Pixel | JPEG | Avg. |
| No_adaptation | 95.5 | 94.9 | 95.4 | 84.9 | 91.7 | 86.6 | 77.1 | 84.1 | 79.4 | 77.3 | 44.3 | 95.8 | 85.2 | 77.4 | 66.8 | 82.4 |
| TENT | 77.4 | 75.2 | 75.8 | 78.0 | 79.9 | 65.3 | 54.2 | 55.9 | 62.2 | 45.4 | 34.5 | 88.4 | 48.9 | 45.5 | 53.2 | 62.6 |
| Bn_adapt | 88.4 | 87.8 | 87.7 | 88.4 | 88.0 | 78.6 | 64.4 | 68.1 | 69.4 | 55.3 | 36.8 | 89.3 | 59.2 | 57.3 | 67.7 | 72.4 |
| SHOT | 77.6 | 74.9 | 76.0 | 80.4 | 80.9 | 63.5 | 53.1 | 54.0 | 61.7 | 45.1 | **34.0** | 95.3 | 47.9 | 45.0 | 52.6 | 62.8 |
| ConjugatePL | 76.5 | 74.1 | 75.4 | 77.4 | 78.4 | 64.8 | 54.1 | 55.5 | 61.5 | 45.2 | 34.4 | 85.6 | 48.6 | 45.3 | 52.9 | 62.0 |
| SAR | 75.2 | 74.4 | 73.9 | 77.5 | **77.7** | 63.5 | 54.0 | 55.6 | 60.5 | 45.4 | 34.5 | **76.7** | 48.9 | 45.5 | 52.9 | 61.1 |
| AEA (ours) | **73.8** | **73.2** | **72.7** | 75.8 | 79.2 | **59.7** | **51.9** | **52.7** | 58.6 | 44.0 | 34.3 | 90.5 | **46.6** | **43.3** | 50.5 | **60.5** |

Table 10: Classification errors (%, ↓) on ImageNet-C (10K target samples) with the highest severity level 5 for ResNet-50 backbone.

| | Noise | | | Blur | | | | Weather | | | | Digital | | | | |
|---|---|---|---|---|---|---|---|---|---|---|---|---|---|---|---|---|
| Method | Gauss. | Shot | Impul. | Defoc. | Glass | Motion | Zoom | Snow | Frost | Fog | Brit. | Contr. | Elastic | Pixel | JPEG | Avg. |
| No_adaptation | 95.5 | 94.8 | 95.4 | 84.8 | 91.7 | 86.2 | 76.8 | 84.0 | 79.3 | 77.1 | 44.4 | 95.6 | 85.3 | 77.1 | 66.5 | 82.3 |
| TENT | 83.6 | 82.2 | 82.9 | 84.0 | 84.1 | 73.0 | 59.5 | 62.4 | 65.2 | 49.6 | 35.6 | 85.9 | 54.4 | 50.3 | 60.0 | 67.5 |
| Bn_adapt | 88.4 | 87.7 | 87.6 | 88.4 | 87.9 | 78.9 | 64.5 | 68.1 | 69.0 | 55.0 | 37.0 | 89.2 | 59.3 | 57.2 | 67.5 | 72.4 |
| SHOT | 81.1 | 79.2 | 80.2 | 84.2 | 83.8 | 70.2 | 57.3 | 58.6 | 63.7 | 47.8 | 35.1 | 92.0 | 51.9 | 48.6 | 56.7 | 66.0 |
| ConjugatePL | 84.0 | 82.6 | 83.3 | 84.5 | 84.4 | 73.3 | 59.7 | 62.6 | 65.4 | 49.7 | 35.6 | 85.6 | 54.7 | 50.5 | 60.1 | 67.7 |
| SAR | 83.7 | 83.7 | 82.9 | 85.3 | 84.9 | 72.8 | 59.9 | 62.5 | 65.2 | 50.0 | 35.8 | **83.3** | 55.0 | 50.8 | 60.0 | 67.7 |
| AEA (ours) | **77.2** | **75.8** | **76.5** | 78.8 | 79.8 | **63.6** | **54.2** | **54.6** | 59.9 | 45.6 | **34.7** | 83.5 | **48.5** | **45.1** | 52.6 | **62.0** |

## B.2 Performance under different backbones

In table 11, we report experimental results for different backbone architectures of vision transformer (ViT) family (Dosovitskiy et al., 2021). Our method can be easily applied to various model architectures and, as shown in the results, consistently achieves superior TTA performance compared to other comparable baselines.

| Method | Gauss. | Shot | Impul. | Defoc. | Glass | Motion | Zoom | Snow | Frost | Fog | Brit. | Contr. | Elastic | Pixel | JPEG | Avg. |
|--------|--------|------|--------|--------|-------|--------|------|------|-------|-----|-------|--------|---------|-------|------|------|
| No_adapt. | 33.5 | 28.3 | 17.8 | 5.8 | 22.6 | 10.6 | 4.9 | 5.5 | 7.3 | 12.9 | 2.9 | 10.0 | 12.7 | 24.4 | 15.5 | 14.3 |
| TENT | 24.2 | 18.9 | 14.0 | 4.7 | 16.6 | 7.8 | 3.9 | 5.0 | 6.3 | 9.0 | 2.7 | 5.8 | 10.3 | 8.6 | 13.4 | 10.1 |
| SAR | 28.1 | 19.8 | 14.4 | 4.8 | 16.6 | 8.0 | 4.0 | 5.1 | 6.4 | 9.5 | 2.7 | 6.0 | 10.4 | 8.8 | 13.5 | 10.5 |
| **AEA(ours)** | 20.3 | 16.3 | 11.5 | 4.7 | 15.0 | 6.6 | 3.7 | 5.1 | 5.3 | 7.0 | 2.7 | 3.9 | 10.1 | 6.9 | 13.6 | **8.8** |
| Method | Gauss. | Shot | Impul. | Defoc. | Glass | Motion | Zoom | Snow | Frost | Fog | Brit. | Contr. | Elastic | Pixel | JPEG | Avg. |
| No_adaptation | 65.1 | 66.3 | 63.6 | 67.6 | 77.8 | 63.4 | 68.9 | 77.8 | 73.3 | 46.8 | 38.8 | 49.7 | 68.0 | 45.5 | 44.1 | 61.1 |
| TENT | 48.2 | 47.4 | 47.1 | 47.7 | 55.5 | 45.0 | 52.6 | 86.5 | 84.3 | 34.2 | 27.2 | 35.0 | 62.5 | 33.4 | 35.6 | 49.5 |
| SAR | 48.2 | 47.2 | 47.1 | 47.6 | 53.8 | 45.1 | 51.7 | 80.2 | 68.6 | 33.9 | 27.9 | 35.4 | 49.4 | 33.9 | 35.7 | 47.0 |
| **AEA(ours)** | 45.9 | 44.9 | 44.4 | 44.5 | 48.1 | 41.4 | 50.7 | 88.6 | 74.3 | 30.8 | 26.3 | 35.5 | 40.9 | 29.9 | 33.1 | **45.3** |

Table 11: Classification errors (%, ↓) on CIFAR10-C for ViT_small_16 backbone (top) and on ImageNet-C for ViT_base_16 backbone (bottom).

## B.3 Performance on Segmentation task

For further justification, we evaluate our method on the segmentation TTA task in Table 12. We report the segmentation performance (i.e., MIoU(%)) on the Cityscapes dataset Cordts et al. (2016), following the experimental setup of Volpi et al. (2022). More specifically, given a pre-trained segmentation model on GTA-5 dataset Richter et al. (2016), we perform online test-time adaptation to adapt the pre-trained model to target sequences composed of target samples from each scene (i.e., city) in Cityscapes dataset. In Table 12, we report the results for five different scenes and other experimental details (e.g., segmentation model, batch size, etc.) are consistent with Volpi et al. (2022). Also, we use the same hyperparameters in our main results for AEA evaluation. The results show that our AEA still achieves competitive performance in the segmentation TTA task, demonstrating its effectiveness and applicability across various tasks.

| Target sequence | Seq.#1 | Seq.#2 | Seq.#3 | Seq.#4 | Seq.#5 | Avg. |
|-----------------|--------|--------|--------|--------|--------|------|
| No adaptation | 42.8 | 41.9 | 47.6 | 46.3 | 47.9 | 45.3 |
| BN adaptation | 43.3 | 42.7 | 48.1 | 47.0 | 48.5 | 45.9 |
| TENT | 43.7 | 43.6 | 48.8 | 47.8 | 49.1 | 46.6 |
| **AEA (ours)** | **44.7** | **44.9** | **50.3** | **48.3** | **49.3** | **47.5** |

Table 12: Evaluation under the TTA segmentation task, following the experimental setup of Volpi et al. (2022). We report the segmentation performance (i.e., MIoU(%, ↑)) on the Cityscapes dataset Cordts et al. (2016). We adapt a pre-trained segmentation model on GTA-5 dataset to five different target sequences from Cityscapes: #1) aachen, #2) dusseldorf, #3) erfurt, #4) monchengladbach and #5) ulm.

## B.4 Performance under source-free domain adaptation setting

Our AEA can also be applied to the source-free domain adaptation (SFDA) task and we present the results in Table 13. The main difference between SFDA and TTA is that SFDA allows access to the entire target samples for model adaptation, whereas TTA should conduct adaptation over a continuous stream of target batches. To demonstrate the effectiveness of our AEA, we measured performance by varying the number of target samples for adaptation in the SFDA setting. Specifically, we first adapt a pre-trained model on the given target samples, and subsequently, the performance is evaluated over the entire target samples. The results show that our method still achieves competitive performance in the SFDA setting, with a significant performance improvement when the number of target samples is small. For example, given only 30% of the entire target samples (for adaptation), our AEA outperforms SHOT by 6.9% and even surpasses the performance of SHOT adapted on the full dataset. These findings support the advantages of our method in accelerating adaptation and suggest its applicability in other source-free scenarios.

|  | 10% | 20% | 30% | 40% | 50% | 60% | 70% | 80% | 90% | 100% |
|---|---|---|---|---|---|---|---|---|---|---|
| No adaptation | 26.9 | 26.9 | 26.9 | 26.9 | 26.9 | 26.9 | 26.9 | 26.9 | 26.9 | 26.9 |
| TENT | 61.6 | 62.9 | 64.3 | 64.9 | 66.6 | 67.5 | 68.7 | 69.1 | 70.1 | 70.5 |
| SHOT | 63.2 | 67.8 | 70.2 | 71.3 | 72.6 | 73.3 | 74.3 | 74.5 | 75.5 | 75.2 |
| **AEA (ours)** | **67.7** | **72.9** | **77.1** | **77.4** | **77.6** | **77.6** | **77.8** | **77.7** | **77.2** | **76.8** |

Table 13: Performance under SFDA setting. We report classification accuracy ($\%, \uparrow$) with varying the sample sizes (in column) for model adaptation. Experiments are conducted on CIFAR10-C (gaussian noise with severity level 5).

### B.5 ABLATION STUDY ON UPDATING MODEL PARAMETERS

We have further conducted an additional ablation study on the proposed components in our AEA. In Table 14, we show the results (i.e., classification errors) of updating all the parameters in the feature extractor, along with the results of updating only the batch normalization parameters, which is denoted by 'Batch Normalization'. As can be seen, our proposed loss functions, i.e., $\mathcal{L}_{SFEA}$ and $\mathcal{L}_{LCS}$, individually contribute to the performance improvement compared to naive EM baselines. This indicates that our approach works effectively regardless of which parameters are updated.

Table 14: Ablation study for each component of AEA. The results show that each component (i.e., $\mathcal{L}_{SFEA}$, $\mathcal{L}_{LCS}$) of our method individually contributes to the performance improvement in TTA.

| Updated parameters | Methods | CIFAR10-C |
|---|---|---|
|  | No adaptation | 44.4 |
| Batch Normalization | EM | 24.0 |
|  | EM + SFEA | 20.7 |
|  | **EM + SFEA + LCS** | **19.5** |
| Feature Extractor | EM | 22.6 |
|  | EM + SFEA | 21.7 |
|  | **EM + SFEA + LCS** | **20.6** |

### B.6 CHOICE OF HYPER-PARAMETERS $\lambda_1, \lambda_2$

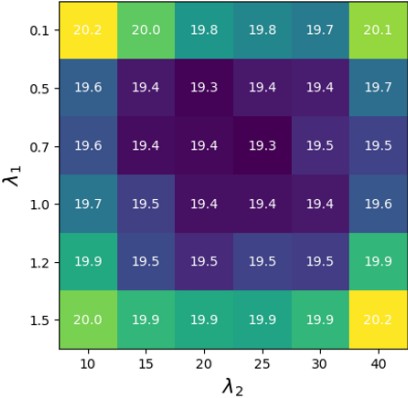

Figure 11: Classification errors ($\%, \downarrow$) on CIFAR10-C with varying our main hyper-parameters (i.e., $\lambda_1, \lambda_2$).

To provide a further insight into the choice of hyper-parameters, we have varied the main hyper-parameters (i.e., $\lambda_1, \lambda_2$) across a wide range and reported the corresponding results in the Fig. 11. Our findings demonstrate that various combinations of $\lambda_1, \lambda_2$ still can yield robust and comparable performances, with the best results when $\lambda_1$ is around 1.0 and $\lambda_2$ is around 25. We note that these results are consistent with the hyper-parameter analysis we provided in Fig. 6 of the main paper.

## B.7 Time Complexity and Memory Usage

In this section, we compare the time complexity and memory usage between our method and baselines. In Fig. 12, we measured performance changes according to the time required for adaptation on CIFAR10-C. Also, in Table 15, we report GPU memory usages (MiB) per adaptation batch during TTA. As can be seen from the results, our scheme achieves high performance more rapidly and requires almost the same amount of computations as TENT, which demonstrates that our proposed method requires only negligible additional costs. Since our proposed SFEA and LCS losses are simply computed using only output logits and the weights of the last classifier, our proposed method requires only negligible additional costs compared to the conventional EM loss. Therefore, our AEA can be practically utilized in various scenarios and applications compared to the other promising baselines.

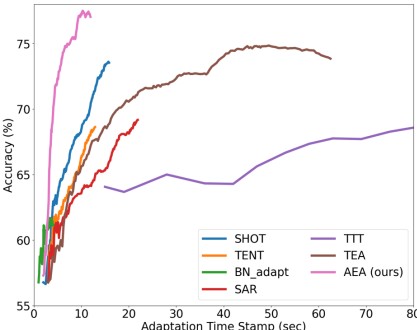

Figure 12: Classification accuracy (%, ↑) with respect to the adaptation time stamp on CIFAR10-C.

| Method | CIFAR10-C/100-C | TinyImageNet-C |
|---|---|---|
| TENT | 1,759 | 9,362 |
| SHOT | 1,829 | 9,973 |
| SAR | 1,761 | 9,367 |
| TEA | 1,919 | 16,449 |
| **AEA (ours)** | **1,761** | **9,367** |

Table 15: GPU memory usage (MiB) per batch during adaptation.

## B.8 Results under the continual TTA settings

In Table 16, we have compared our AEA with several promising baselines under the continual TTA scenario. The key consideration under the continual TTA is to adapt the model persistently to changing domains while mitigating error accumulation or catastrophic forgetting (Wang et al., 2022). We note that our AEA does not primarily target the continual setting and thus has no additional components for those purposes. To compensate for this, we can adopt a simple model restoration technique (e.g., stochastic restoration in Wang et al. (2022)) which stochastically resets the adapted model to its initial state. In Table 16, we report our AEA's results under continual setting with stochastic restoration technique. For a fair comparison, we also apply the same technique to baseline that do not include additional components for continual settings, such as TENT (Wang et al., 2021). Interestingly, the results confirms that although we simply restore the model to its initial state stochastically, our method can also achieve competitive performance in the continual TTA setting. We attribute this to our AEA's capability of effectively accelerating model adaptation within a few batches despite model restorations via adaptive energy alignment scheme. This enables the model to rapidly adapt to new target domains and achieve competent OOD adaptation performance. In this regard, we argue that our AEA's acceleration capability could be another beneficial approach to tackle the continual TTA settings.

Table 16: Classification errors (%, ↓) on CIFAR100-C with the highest severity level 5 for ResNext-29 backbone. The experiment is conducted under the continual TTA setup following Wang et al. (2022), and the domain shifts happen from left (Gauss.) to right (JPEG). We report the averaged results over 3 different runs.

| Method | Noise | | | Blur | | | | Weather | | | | Digital | | | | Avg. |
|---|---|---|---|---|---|---|---|---|---|---|---|---|---|---|---|---|
| | Gauss. | Shot | Impul. | Defoc. | Glass | Motion | Zoom | Snow | Frost | Fog | Brit. | Contr. | Elastic | Pixel | JPEG | |
| No adaptation | 73.0 | 68.0 | 39.4 | 29.4 | 54.1 | 30.8 | 28.8 | 39.5 | 45.8 | 50.3 | 29.5 | 55.1 | 37.2 | 74.7 | 41.2 | 46.5 |
| BN adaptation (Schneider et al., 2020) | 42.3 | 40.8 | 43.2 | 27.7 | 41.8 | 29.8 | 27.9 | 35.0 | 34.8 | 41.7 | 26.4 | 30.2 | 35.6 | 33.2 | 41.2 | 35.4 |
| TENT (Wang et al., 2021) | 37.2 | 34.3 | 36.3 | 27.3 | 39.4 | 30.1 | 27.1 | 32.4 | 33.1 | 35.8 | 24.8 | 28.0 | 35.2 | 30.8 | 39.1 | 32.7 |
| EATA (Niu et al., 2022) | 37.2 | 33.1 | 36.2 | 27.8 | 37.9 | 29.7 | 26.9 | 32.7 | 31.3 | 35.3 | 26.9 | 28.7 | 33.4 | 29.8 | 37.4 | 32.3 |
| SAR (Niu et al., 2023) | 40.5 | 34.9 | 37.1 | 25.9 | 37.2 | 28.0 | 25.6 | 31.8 | 30.8 | 35.9 | 25.2 | 28.1 | 32.1 | 29.2 | 37.3 | 32.0 |
| RoTTA (Yuan et al., 2023) | 49.1 | 45.0 | 45.5 | 30.2 | 42.6 | 29.5 | 26.0 | 32.3 | 30.6 | 37.6 | 24.7 | 29.2 | 32.7 | 30.4 | 36.9 | 34.8 |
| COTTA (Wang et al., 2022) | 40.6 | 38.0 | 39.9 | 27.2 | 38.2 | 28.5 | 26.5 | 33.3 | 32.2 | 40.6 | 25.1 | 26.9 | 32.3 | 28.3 | 33.8 | 32.8 |
| AEA (ours) | 36.2 | 32.8 | 32.8 | 25.3 | 36.3 | 27.5 | 24.8 | 29.7 | 29.0 | 31.8 | 24.0 | 25.3 | 32.2 | 28.1 | 35.8 | 30.1 |

## B.9 RESULTS WITH STANDARD DEVIATIONS

In Tables 17-19, we present the online TTA results on CIFAR10-C, CIFAR100-C, and TinyImageNet-C with standard deviations, respectively. Each method is conducted with three different random seeds, and we report the mean and standard deviation of three runs for each method. Our AEA, which is fundamentally based on the EM loss, shows similar or lower standard deviations compared to TENT. The results still confirm that our method consistently achieves superior TTA performance through our energy alignment approach.

Table 17: Classification errors (%, ↓) with standard deviations on CIFAR10-C with the highest severity level 5 for ResNet-26 backbone.

| Method | Noise | | | Blur | | | | Weather | | | | Digital | | | | Avg. |
|---|---|---|---|---|---|---|---|---|---|---|---|---|---|---|---|---|
| | Gauss. | Shot | Impul. | Defoc. | Glass | Motion | Zoom | Snow | Frost | Fog | Brit. | Contr. | Elastic | Pixel | JPEG | |
| No adaptation | 73.1 | 66.8 | 69.4 | 40.2 | 51.9 | 37.9 | 38.1 | 25.5 | 39.3 | 39.4 | 10.1 | 55.2 | 26.7 | 61.5 | 31.0 | 44.4 |
| | (±0.00) | (±0.00) | (±0.00) | (±0.00) | (±0.00) | (±0.00) | (±0.00) | (±0.00) | (±0.00) | (±0.00) | (±0.00) | (±0.00) | (±0.00) | (±0.00) | (±0.00) | (±0.00) |
| TENT | 32.7 | 30.2 | 39.1 | 15.7 | 35.9 | 18.2 | 16.1 | 21.8 | 23.1 | 19.4 | 12.7 | 16.0 | 26.1 | 22.3 | 30.4 | 24.0 |
| | (±0.52) | (±0.34) | (±0.10) | (±0.16) | (±0.58) | (±0.13) | (±0.28) | (±0.10) | (±0.30) | (±0.15) | (±0.29) | (±0.43) | (±0.56) | (±0.42) | (±0.50) | (±0.33) |
| BN adaptation | 39.1 | 36.6 | 46.1 | 17.2 | 41.1 | 19.8 | 18.0 | 25.1 | 25.5 | 21.0 | 14.1 | 17.6 | 28.9 | 26.6 | 35.5 | 27.5 |
| | (±0.11) | (±0.13) | (±0.07) | (±0.18) | (±0.33) | (±0.13) | (±0.10) | (±0.12) | (±0.13) | (±0.07) | (±0.18) | (±0.23) | (±0.23) | (±0.11) | | (±0.14) |
| SHOT | 29.5 | 26.5 | 35.4 | 14.6 | 33.7 | 17.0 | 14.6 | 19.8 | 22.2 | 17.8 | 11.8 | 16.2 | 24.6 | 20.4 | 26.9 | 22.1 |
| | (±0.76) | (±0.40) | (±0.52) | (±0.04) | (±1.01) | (±0.08) | (±0.49) | (±0.34) | (±0.84) | (±0.18) | (±0.49) | (±0.99) | (±0.32) | (±0.51) | (±0.58) | (±0.50) |
| T3A | 65.3 | 59.6 | 65.5 | 37.0 | 47.7 | 34.5 | 34.0 | 24.8 | 36.6 | 34.6 | 10.1 | 50.0 | 25.1 | 51.8 | 29.8 | 40.4 |
| | (±0.22) | (±0.12) | (±0.15) | (±0.05) | (±0.02) | (±0.16) | (±0.05) | (±0.05) | (±0.12) | (±0.15) | (±0.02) | (±0.25) | (±0.03) | (±0.08) | (±0.08) | (±0.10) |
| TTT | 24.9 | 23.0 | 30.2 | 13.5 | 34.6 | 20.4 | 15.9 | 19.3 | 17.9 | 14.1 | 9.4 | 26.4 | 23.7 | 16.0 | 23.8 | 20.9 |
| | (±0.74) | (±0.04) | (±0.13) | (±0.19) | (±0.32) | (±0.23) | (±0.04) | (±0.05) | (±0.12) | (±0.15) | (±0.08) | (±0.42) | (±0.26) | (±0.17) | (±0.17) | (±0.22) |
| NOTE | 48.8 | 42.5 | 47.5 | 25.0 | 40.9 | 24.4 | 23.7 | 20.3 | 23.5 | 22.8 | 9.1 | 31.6 | 24.7 | 41.9 | 29.2 | 30.4 |
| | (±0.57) | (±0.29) | (±0.42) | (±0.50) | (±0.25) | (±0.16) | (±0.12) | (±0.16) | (±0.23) | (±0.16) | (±0.16) | (±0.29) | (±0.18) | (±0.14) | (±0.19) | (±0.25) |
| Conjugate PL | 32.7 | 30.1 | 39.2 | 15.8 | 35.8 | 18.2 | 16.1 | 21.8 | 23.1 | 19.4 | 12.7 | 16.1 | 26.1 | 22.3 | 30.4 | 24.0 |
| | (±0.45) | (±0.36) | (±0.12) | (±0.19) | (±0.58) | (±0.13) | (±0.28) | (±0.10) | (±0.32) | (±0.14) | (±0.29) | (±0.44) | (±0.53) | (±0.45) | (±0.48) | (±0.32) |
| EATA | 38.7 | 36.2 | 46.0 | 17.1 | 40.8 | 19.8 | 17.9 | 24.9 | 25.4 | 20.9 | 14.1 | 17.6 | 28.7 | 26.4 | 35.3 | 27.3 |
| | (±0.17) | (±0.40) | (±0.19) | (±0.21) | (±0.26) | (±0.06) | (±0.12) | (±0.22) | (±0.09) | (±0.11) | (±0.03) | (±0.20) | (±0.34) | (±0.23) | (±0.17) | (±0.19) |
| SAR | 32.9 | 30.6 | 39.5 | 15.9 | 36.3 | 18.3 | 16.3 | 22.1 | 23.2 | 19.3 | 12.7 | 16.5 | 26.2 | 22.8 | 30.6 | 24.2 |
| | (±0.13) | (±0.22) | (±0.28) | (±0.20) | (±0.31) | (±0.08) | (±0.12) | (±0.13) | (±0.06) | (±0.23) | (±0.12) | (±0.17) | (±0.21) | (±0.26) | (±0.52) | (±0.20) |
| TEA | 27.7 | 25.5 | 34.2 | 15.3 | 34.8 | 18.0 | 15.9 | 20.0 | 20.5 | 17.7 | 12.4 | 16.4 | 25.7 | 19.2 | 26.3 | 22.0 |
| | (±0.27) | (±0.40) | (±0.64) | (±0.22) | (±0.95) | (±0.23) | (±0.45) | (±0.09) | (±0.31) | (±0.19) | (±0.24) | (±0.24) | (±0.12) | (±0.10) | (±0.22) | (±0.31) |
| AEA (ours) | 24.7 | 22.7 | 31.9 | 13.8 | 30.8 | 16.2 | 13.8 | 17.5 | 17.5 | 15.4 | 10.8 | 13.7 | 23.3 | 17.4 | 23.8 | 19.5 |
| | (±0.19) | (±0.05) | (±0.58) | (±0.10) | (±0.86) | (±0.38) | (±0.34) | (±0.20) | (±0.17) | (±0.39) | (±0.20) | (±0.42) | (±0.27) | (±0.12) | (±0.17) | (±0.30) |

## C IMPLEMENTATION DETAILS

In this section, we provide some detailed implementation settings for experiments. Basically, we follow the basic settings of Zhao et al. (2023), including the hyperparameter settings required for each baseline. For all experiments, we use SGD as optimizer and set the learning rate to 0.001, batch size to 64, and the number of adaptation iterations per batch to 1. Also, we adapt the model parameters of the batch normalization (BN) layers similar to Wang et al. (2021). Additionally, there are several hyperparameters that need to be tuned for our AEA, i.e., $\{\lambda_1, \alpha\}$ for the SFEA loss and $\{\lambda_2, C_0\}$ for the LCS loss. For all datasets in the experiment section, we apply the SFEA loss with $\lambda_1$ set to 1.0 and $\alpha$ set to 0.5. Also, for the LCS loss, which depends on the confidence scores of the model, we set the threshold $C_0$ to 0.99 for CIFAR10-C, 0.66 for CIFAR100-C and TinyImageNet-C. Also, $\lambda_2$ is set to 25 for all experiments. To further provide insights into our hyperparameters, we conduct an ablation study for each hyperparameter in Fig. 6 and Sec. B.6. All the experiments are performed with 1×NVIDIA GeForce RTX 3090 and the other details of computational resources (e.g. workers, memory, etc.) are the same with Zhao et al. (2023).

Table 18: Classification errors (%, ↓) with standard deviations on CIFAR100-C with the highest severity level 5 for ResNet-26 backbone.

| Method | Noise Gauss. | Shot | Impul. | Blur Defoc. | Glass | Motion | Zoom | Weather Snow | Frost | Fog | Brit. | Digital Contr. | Elastic | Pixel | JPEG | Avg. |
|---|---|---|---|---|---|---|---|---|---|---|---|---|---|---|---|---|
| No adaptation | 89.3 (±0.00) | 88.3 (±0.00) | 91.0 (±0.00) | 67.2 (±0.00) | 63.5 (±0.00) | 60.8 (±0.00) | 59.6 (±0.00) | 56.1 (±0.00) | 62.3 (±0.00) | 67.6 (±0.00) | 42.7 (±0.00) | 84.4 (±0.00) | 50.8 (±0.00) | 85.5 (±0.00) | 60.9 (±0.00) | 68.7 (±0.00) |
| TENT | 65.5 (±0.18) | 64.7 (±0.12) | 65.1 (±0.33) | 43.9 (±0.14) | 58.1 (±0.17) | 47.0 (±0.21) | 43.6 (±0.29) | 56.2 (±0.37) | 54.1 (±0.43) | 52.3 (±0.27) | 42.9 (±0.43) | 49.5 (±0.30) | 51.4 (±0.39) | 50.7 (±0.15) | 59.6 (±0.10) | 53.6 (±0.26) |
| BN adaptation | 70.5 (±0.20) | 69.9 (±0.13) | 68.8 (±0.33) | 46.6 (±0.15) | 60.8 (±0.33) | 48.8 (±0.10) | 45.9 (±0.13) | 59.0 (±0.33) | 56.8 (±0.09) | 55.1 (±0.35) | 45.5 (±0.19) | 51.2 (±0.30) | 53.5 (±0.31) | 54.8 (±0.03) | 62.8 (±0.14) | 56.7 (±0.21) |
| SHOT | 58.6 (±0.64) | 57.7 (±0.23) | 58.7 (±0.66) | 41.4 (±0.10) | 55.0 (±0.11) | 44.1 (±0.37) | 41.3 (±0.10) | 51.9 (±0.32) | 49.9 (±0.19) | 48.6 (±0.15) | 41.0 (±0.13) | 48.6 (±0.46) | 48.9 (±0.41) | 46.4 (±0.62) | 56.0 (±0.38) | 49.9 (±0.32) |
| T3A | 89.3 (±0.14) | 88.4 (±0.17) | 90.4 (±0.04) | 64.8 (±0.25) | 60.9 (±0.19) | 59.9 (±0.25) | 57.3 (±0.17) | 57.2 (±0.05) | 61.2 (±0.12) | 65.3 (±0.34) | 43.2 (±0.02) | 82.5 (±0.01) | 50.0 (±0.06) | 82.9 (±0.07) | 60.4 (±0.29) | 67.6 (±0.14) |
| TTT | 63.7 (±0.14) | 63.2 (±0.09) | 65.1 (±0.29) | 43.9 (±0.20) | 57.2 (±0.03) | 49.9 (±0.20) | 43.4 (±0.13) | 54.1 (±0.28) | 50.8 (±0.55) | 49.7 (±0.10) | 38.7 (±0.17) | 70.2 (±0.27) | 49.7 (±0.22) | 45.7 (±0.29) | 56.1 (±0.37) | 53.4 (±0.22) |
| NOTE | 76.4 (±0.29) | 74.6 (±0.04) | 74.5 (±0.17) | 53.9 (±0.25) | 57.6 (±0.22) | 50.7 (±0.27) | 47.9 (±0.18) | 52.7 (±0.27) | 52.3 (±0.31) | 56.7 (±0.34) | 38.6 (±0.17) | 67.4 (±0.45) | 49.0 (±0.25) | 70.4 (±0.55) | 57.8 (±0.06) | 58.7 (±0.25) |
| Conjugate PL | 65.6 (±0.16) | 64.7 (±0.14) | 65.1 (±0.34) | 43.9 (±0.15) | 58.1 (±0.17) | 47.0 (±0.22) | 43.6 (±0.31) | 56.2 (±0.38) | 54.1 (±0.45) | 52.3 (±0.25) | 42.9 (±0.40) | 49.5 (±0.28) | 51.4 (±0.38) | 50.7 (±0.18) | 59.6 (±0.08) | 53.6 (±0.26) |
| EATA | 68.0 (±2.67) | 66.2 (±2.50) | 71.5 (±1.38) | 46.0 (±0.31) | 64.7 (±1.13) | 49.3 (±1.21) | 46.0 (±1.33) | 56.7 (±1.03) | 57.2 (±1.73) | 53.8 (±0.13) | 44.1 (±0.69) | 51.9 (±1.96) | 55.4 (±1.36) | 52.1 (±3.23) | 62.2 (±0.99) | 56.3 (±1.44) |
| SAR | 65.8 (±0.08) | 64.9 (±0.32) | 65.3 (±0.26) | 44.2 (±0.17) | 58.2 (±0.26) | 47.1 (±0.06) | 43.8 (±0.46) | 56.4 (±0.35) | 54.4 (±0.31) | 52.5 (±0.33) | 43.0 (±0.32) | 49.3 (±0.37) | 51.4 (±0.40) | 50.8 (±0.04) | 59.7 (±0.28) | 53.8 (±0.27) |
| TEA | 64.0 (±0.43) | 63.3 (±0.09) | 64.2 (±0.57) | 45.1 (±0.29) | 59.0 (±0.23) | 48.4 (±0.23) | 45.4 (±0.43) | 56.5 (±0.28) | 55.1 (±0.44) | 52.5 (±0.45) | 43.3 (±0.27) | 53.4 (±0.70) | 52.7 (±0.27) | 50.1 (±0.10) | 60.0 (±0.17) | 54.2 (±0.33) |
| AEA (ours) | 58.2 (±0.57) | 58.8 (±0.27) | 59.0 (±0.24) | 40.9 (±0.20) | 55.0 (±0.15) | 43.9 (±0.44) | 40.5 (±0.31) | 51.3 (±0.05) | 49.0 (±0.31) | 47.4 (±0.27) | 39.4 (±0.16) | 44.1 (±0.51) | 48.4 (±0.30) | 44.3 (±0.48) | 54.8 (±0.25) | 49.0 (±0.30) |

Table 19: Classification errors (%, ↓) with standard deviations on TinyImageNet-C with the highest severity level 5 for ResNet-50 backbone.

| Method | Noise Gauss. | Shot | Impul. | Blur Defoc. | Glass | Motion | Zoom | Weather Snow | Frost | Fog | Brit. | Digital Contr. | Elastic | Pixel | JPEG | Avg. |
|---|---|---|---|---|---|---|---|---|---|---|---|---|---|---|---|---|
| No adaptation | 96.6 (±0.00) | 95.1 (±0.00) | 97.2 (±0.00) | 92.5 (±0.00) | 92.2 (±0.00) | 77.8 (±0.00) | 78.5 (±0.00) | 81.9 (±0.00) | 78.1 (±0.00) | 89.5 (±0.00) | 77.8 (±0.00) | 98.3 (±0.00) | 69.5 (±0.00) | 71.9 (±0.00) | 55.6 (±0.00) | 83.5 (±0.00) |
| SHOT | 64.5 (±0.15) | 62.8 (±0.25) | 70.6 (±0.42) | 61.9 (±0.17) | 73.5 (±0.20) | 54.5 (±0.37) | 53.8 (±0.48) | 62.1 (±0.34) | 61.1 (±0.37) | 65.2 (±0.25) | 56.2 (±0.08) | 89.6 (±1.70) | 55.4 (±0.25) | 50.8 (±0.32) | 51.8 (±0.25) | 62.2 (±0.37) |
| TENT | 66.5 (±0.32) | 64.3 (±0.46) | 72.9 (±0.16) | 63.2 (±0.30) | 75.1 (±0.13) | 55.5 (±0.20) | 54.8 (±0.28) | 63.6 (±0.43) | 61.9 (±0.16) | 67.6 (±0.28) | 57.6 (±0.05) | 86.2 (±0.09) | 56.6 (±0.33) | 51.5 (±0.29) | 52.5 (±0.15) | 63.3 (±0.24) |
| Bn_adapt | 68.8 (±0.32) | 66.7 (±0.43) | 75.7 (±0.27) | 65.0 (±0.27) | 77.1 (±0.18) | 56.4 (±0.30) | 55.8 (±0.41) | 64.9 (±0.16) | 63.4 (±0.27) | 71.0 (±0.17) | 59.0 (±0.08) | 86.5 (±0.12) | 57.7 (±0.26) | 52.2 (±0.26) | 53.3 (±0.19) | 64.9 (±0.24) |
| T3A | 96.6 (±0.11) | 95.0 (±0.09) | 97.3 (±0.06) | 92.7 (±0.08) | 92.1 (±0.12) | 77.3 (±0.27) | 78.2 (±0.02) | 82.3 (±0.13) | 78.2 (±0.13) | 89.9 (±0.08) | 77.1 (±0.18) | 98.5 (±0.10) | 68.9 (±0.24) | 70.3 (±0.09) | 55.8 (±0.08) | 83.4 (±0.11) |
| NOTE | 83.4 (±0.18) | 80.4 (±0.14) | 86.4 (±0.26) | 81.9 (±0.05) | 85.7 (±0.15) | 65.4 (±0.23) | 64.1 (±0.30) | 69.6 (±0.08) | 67.4 (±0.08) | 79.8 (±0.06) | 64.2 (±0.21) | 96.0 (±0.15) | 61.2 (±0.24) | 56.0 (±0.24) | 53.1 (±0.11) | 73.0 (±0.16) |
| SAR | 67.2 (±0.22) | 65.2 (±0.23) | 73.6 (±0.17) | 63.9 (±0.28) | 75.9 (±0.18) | 55.8 (±0.20) | 55.2 (±0.32) | 64.1 (±0.28) | 62.6 (±0.27) | 68.4 (±0.24) | 58.0 (±0.06) | 86.0 (±0.13) | 57.1 (±0.27) | 51.8 (±0.34) | 52.9 (±0.25) | 63.8 (±0.23) |
| ConjugatePL | 67.4 (±0.17) | 65.3 (±0.22) | 74.0 (±0.12) | 63.9 (±0.28) | 76.1 (±0.22) | 55.8 (±0.23) | 55.3 (±0.29) | 64.3 (±0.30) | 62.7 (±0.27) | 68.6 (±0.21) | 58.1 (±0.07) | 86.2 (±0.04) | 57.1 (±0.26) | 51.8 (±0.37) | 53.0 (±0.26) | 64.0 (±0.22) |
| EATA | 65.1 (±0.27) | 63.6 (±0.46) | 69.7 (±0.28) | 62.3 (±0.44) | 74.2 (±0.54) | 55.3 (±0.09) | 54.1 (±0.21) | 61.3 (±0.28) | 61.0 (±0.14) | 63.9 (±0.49) | 55.4 (±0.34) | 90.8 (±0.28) | 56.1 (±0.40) | 51.1 (±0.62) | 52.2 (±0.10) | 62.4 (±0.33) |
| TEA | 66.1 (±0.26) | 64.6 (±0.24) | 72.3 (±0.15) | 63.2 (±0.13) | 75.1 (±0.10) | 55.8 (±0.20) | 54.9 (±0.24) | 63.1 (±0.47) | 61.2 (±0.08) | 65.9 (±0.21) | 56.9 (±0.30) | 85.3 (±0.14) | 56.9 (±0.17) | 51.8 (±0.08) | 52.9 (±0.15) | 63.1 (±0.19) |
| AEA (ours) | 63.7 (±0.13) | 62.1 (±0.29) | 67.9 (±0.31) | 60.9 (±0.20) | 72.8 (±0.32) | 53.8 (±0.06) | 52.9 (±0.16) | 60.1 (±0.16) | 58.9 (±0.22) | 61.9 (±0.21) | 54.2 (±0.18) | 90.3 (±0.88) | 54.5 (±0.35) | 49.8 (±0.25) | 51.1 (±0.08) | 61.0 (±0.25) |

# D LIMITATIONS OF OUR WORK

As our AEA is primarily designed for the online TTA setting, it does not include mechanisms to mitigate error accumulation or catastrophic forgetting issues for continual setting. While AEA still achieves competitive performance in continual TTA due to its acceleration capability, a promising future direction would be to enhance its applicability to continual setups by integrating model restoration strategies (e.g., stochastic restoration in Wang et al. (2022)). We hope that our study, which effectively accelerates model adaptation through adaptive energy alignment, will contribute to making TTA more practical and widely applicable across various real-world scenarios.

