# OpenReview forum: "Adaptive Energy Alignment for Accelerating Test-Time Adaptation"
_ICLR.cc/2025/Conference — ICLR 2025 Poster_

### Official Review · Reviewer_w8pg · 2024-10-20

**Soundness:** 3
**Presentation:** 3
**Contribution:** 3
**Rating:** 8
**Confidence:** 3

**Summary:**

This paper proposes Adaptive Energy Alignment (AEA) for fast test-time adaptation (TTA) in out-of-domain scenarios. It reinterprets entropy minimization (EM) loss and shows that its conflicting energy-based components can hinder adaptation. AEA reduces the energy gap between source and target domains by aligning target sample energy levels with familiar source domains and matching class-wise correlations. The approach achieves efficient TTA with minimal computational overhead, demonstrating strong results on datasets like CIFAR10-C, CIFAR100-C, and TinyImageNet-C.

**Strengths:**

*	This paper is well-written, well-organized, and easy to follow.
*	The insight of leveraging the direction of energy alignment with the guidance of the structural relations between different classes is convince, and provide a new solution for accelerating test-time adaptation
*	The proposed SFEA and LCS loss is novel to me, effectively reduce the energy gap and aligns the class-wise correlation across source and target domains.
*	Overall, the quality of the paper is commendable. The authors have conducted a thorough comparison experiments to examine the effectiveness of the proposed method. Additionally, they have made the code and datasets available in the supplementary material.

**Weaknesses:**

*	While the proposed AEA significantly accelerates test-time adaptation (as observed in Figure 1c), the authors provide insufficient discussion on how AEA achieves this acceleration. Further analysis and discussion are needed.
*	I noticed that in TEA, the backbone for CIFAR10-C is WRN-28-10, whereas ResNet-26 is used in this work. If the experimental results are reproduced using publicly available code, please provide the relevant details in the table caption.

**Questions:**

Please refer to the weakness part.

---
I am open to discussing these points with the authors during the response period. If the concerns and questions are adequately addressed, I will consider raising my score.

---

> ### Author Response · Authors · 2024-11-22
>
> ### **Further analysis and discussion on AEA's TTA acceleration**
> >
> >Thank you for your valuable suggestions. **Our AEA facilitates model adaptation by aggressively reducing the energy gap from the early online batches, which allows rapid alignment of the target feature with that of the source domain.** Notably, previous studies have already suggested that the feature alignment between the source and target domains can alleviate the domain gap and achieve better adaptation. Consequently, as we discussed in Sec. 4.3, our AEA turns out to promote fast feature alignment, leading to improved performance in the early adaptation batches (i.e., acceleration). We have discussed these points in the subsection titled “Why does energy alignment lead to better adaptation?” in Sect. 4.3, and endeavored to describe them as thoroughly as possible in Fig. 7.
> >
> >&nbsp;
> >
> >To further demonstrate that **our energy reduction method indeed aligns the true source and target features rapidly,** in Table 1, we measured **the true distance between source features and target features during TTA adaptation.** Specifically, for each adaptation stage, we (1) estimate the source/target feature means for each ground-truth class, (2) measure the Euclidean distance between the feature means for each class, and (3) take the average across all classes. These measurements can indicate how well the adapted model performs feature alignment between the source and target domains. As can be seen, our AEA achieves better alignment by reducing the distance between each feature from 5.18 to 0.98 over 40 adaptation batches, whereas TENT does from 5.18 to 1.76. Based on these findings, we argue that **our adaptive energy alignment approach facilitates more effective feature alignment from the early online batches, accelerating TTA adaptation.** We have supplemented this additional analysis in Sec. A.8 of the Appendix, and we appreciate your comments.
> >
> >&nbsp;
> >
> >- **Table 1. Feature alignment between the source and target domains.**
> >We measure the true distance (i.e, Euclidean distance) between source features and target features during TTA adaptation. Each column represents the number of adaptation batches. Experiments are conducted on CIFAR10-C (gaussian noise with severity level 5).
> >
> >|               | 0    | 10   | 20   | 30   | 40   | 50   | 60   | 70   | 80   | 90   | 100  | 110  | 120  | 130  | 140  | 150  |
> >| ------------- | ---- | ---- | ---- | ---- | ---- | ---- | ---- | ---- | ---- | ---- | ---- | ---- | ---- | ---- | ---- | ---- |
> >| No adaptation | 5.18 | 5.18 | 5.18 | 5.18 | 5.18 | 5.18 | 5.18 | 5.18 | 5.18 | 5.18 | 5.18 | 5.18 | 5.18 | 5.18 | 5.18 | 5.18 |
> >| TENT          | 5.18 | 1.87 | 1.83 | 1.80 | 1.76 | 1.73 | 1.69 | 1.64 | 1.61 | 1.58 | 1.55 | 1.52 | 1.49 | 1.46 | 1.44 | 1.41 |
> >| AEA (ours)    | 5.18 | **1.58** | **1.28** | **1.09** | **0.98** | **0.94** | **0.89** | **0.86** | **0.82** | **0.79** | **0.78** | **0.77** | **0.75** | **0.71** | **0.69** | **0.71** |
>
>
> ### **Relevant details in the main table captions**
> >
> >Thank you for your comments. To ensure a fair comparison, we have reproduced the experimental results of each method following the setup of a recent benchmark [1], which has standardized the TTA setup with consistent configurations (e.g., pre-trained model, learning rate, batch size, etc.). In accordance with the reviewer’s suggestion, we have included the relevant details in the table captions of the revised manuscript.
> >
> >&nbsp;
> >
> >[1] Zhao et al. On pitfalls of test-time adaptation. ICML’23.
>
> We appreciate the reviewer for the constructive feedback, and taking a positive position. We tried to carefully address the raised points. If there are any remaining concerns, we would be grateful for another chance to respond.

---

> > ### Comment · Reviewer_w8pg · 2024-11-29
> >
> > Thank you for your rebuttal. All of my concerns have been addressed. Therefore, I have decided to increase my score to 8.

---

> > > ### Author Response · Authors · 2024-11-30
> > >
> > > We really appreciate having the opportunity to address the reviewer’s concerns. We are grateful for raising the score. Thank you once again for acknowledging our work and providing valuable comments.

---

### Official Review · Reviewer_SzkH · 2024-10-23

**Soundness:** 3
**Presentation:** 3
**Contribution:** 3
**Rating:** 6
**Confidence:** 3

**Summary:**

This paper proposes an Adaptive Energy Alignment (AEA) method that enables fast online test-time adaptation (TTA). The introduced Source-Free Energy Alignment (SFEA) loss strategically aligns the overall energy magnitudes between source and target domains. Additionally, the authors propose a Logit Cosine Similarity (LCS) loss to ensure that class-wise correlations in the target domain align well with those in the source domain during energy alignment. Extensive experiments on three datasets demonstrate the advantages of the proposed AEA method.

**Strengths:**

*	The proposed SFEA loss provide novel insight for TTA, effectively reduce the domain gap.
*	The authors have conducted a thorough ablation study to examine the impact of various components. Additionally, they have made the code and datasets available in the supplementary material.
*	The paper is well-written, well-organized, and easy to read.

**Weaknesses:**

*	While the authors demonstrate the superiority of the proposed method on TTA, it would be interesting to evaluate its performance in a similar scenario, such as source-free domain adaptation (SFDA), where source data is also inaccessible. Additional evaluations could further illustrate the versatility of the proposed method.
*	The authors did not clearly discuss the distinction between online test-time adaptation (OTTA) and TTA. In line 142, they claim to address the OTTA problem, but in other parts of the paper, the main focus appears to be on TTA.
*	A minor issue concerns the punctuation of equations. Eqs. (3), (4), and (9) should include the appropriate punctuation.

**Questions:**

Please refer to the weakness part.

---

> ### Author Response · Authors · 2024-11-22
>
> ### **Evaluation on the other source-free scenarios (e.g., SFDA)**
> >
> >Thank you for your interesting suggestion. **Our AEA can also be applied to the SFDA task.** During the rebuttal period, we conducted additional experiments and present the results in Table 1. The main difference between SFDA and TTA is that SFDA allows access to the entire target samples for model adaptation, whereas TTA should conduct adaptation over a continuous stream of target batches. To demonstrate the effectiveness of our AEA, we measured performance by varying the number of target samples for adaptation in the SFDA setting (but, the performance is evaluated over the entire target samples). The results show that **our method still achieves competitive performance in the SFDA setting, with a significant performance improvement when the number of target samples is small.** For example, given only 30% of the entire target samples (for adaptation), our AEA outperforms SHOT by 6.9% and even surpasses the performance of SHOT adapted on the full dataset. These findings further support the advantages of our method in accelerating adaptation and suggest its applicability in other source-free scenarios.
> >
> >&nbsp;
> >
> >- **Table 1. Performance under SFDA setting.**
> >We report classification accuracy (%, ↑) with varying the sample sizes (in column) for model adaptation. Experiments are conducted on CIFAR10-C (gaussian noise with severity level 5). Other experimental details are provided in Sec. B.4 of Appendix.
> >
> >|               | 10%  | 20%  | 30%  | 40%  | 50%  | 60%  | 70%  | 80%  | 90%  | 100% |
> >| ------------- | ---- | ---- | ---- | ---- | ---- | ---- | ---- | ---- | ---- | ---- |
> >| No adaptation | 26.9 | 26.9 | 26.9 | 26.9 | 26.9 | 26.9 | 26.9 | 26.9 | 26.9 | 26.9 |
> >| TENT          | 61.6 | 62.9 | 64.3 | 64.9 | 66.6 | 67.5 | 68.7 | 69.1 | 70.1 | 70.5 |
> >| SHOT          | 63.2 | 67.8 | 70.2 | 71.3 | 72.6 | 73.3 | 74.3 | 74.5 | 75.5 | 75.2 |
> >| AEA (ours)    | **67.7** | **72.9** | **77.1** | **77.4** | **77.6** | **77.6** | **77.8** | **77.7** | **77.2** | **76.8** |
> >
> >&nbsp;
> >
> >To further confirm the versatility of the proposed method, in Table 2, **we also include results in a continual TTA setting where the target domains continuously change.** In this scenario, we adopted a simple model restoration approach from [1] to prevent error accumulation. We observed that AEA achieves a satisfactory performance even in this challenging scenario, further demonstrating the strong applicability of our approach.
> >
> >&nbsp;
> >
> >- **Table 2. Classification errors (%, ↓) under continual TTA setting on CIFAR100-C.**
> >Experimental details are provided in Sec. B.9 of Appendix.
> >
> >|                           | Gauss. | Shot | Impul. | Defoc. | Glass | Motion | Zoom | Snow | Frost | Fog  | Brit. | Contr. | Elastic | Pixel | JPEG | Avg. |
> >| ------------------------- | ------ | ---- | ------ | ------ | ----- | ------ | ---- | ---- | ----- | ---- | ----- | ------ | ------- | ----- | ---- | ---- |
> >| No adaptation      | 73.0 | 68.0 | 39.4 | 29.4 | 54.1 | 30.8 | 28.8 | 39.5 | 45.8 | 50.3 | 29.5 | 55.1 | 37.2 | 74.7 | 41.2 | 46.5 |
> >| BN adaptation    | 42.3 | 40.8 | 43.2 | 27.7 | 41.8 | 29.8 | 27.9 | 35.0 | 34.8 | 41.7 | 26.4 | 30.2 | 35.6 | 33.2 | 41.2 | 35.4 |
> >| TENT        | 37.2 | 34.3 | 36.3 | 27.3 | 39.4 | 30.1 | 27.1 | 32.4 | 33.1 | 35.8 | 24.8 | 28.0 | 35.2 | 30.8 | 39.1 | 32.7 |
> >| EATA        | 37.2 | 33.1 | 36.2 | 27.8 | 37.9 | 29.7 | 26.9 | 32.7 | 31.3 | 35.3 | 26.9 | 28.7 | 33.4 | 29.8 | 37.4 | 32.3 |
> >| SAR         | 40.5 | 34.9 | 37.1 | 25.9 | 37.2 | 28.0 | 25.6 | 31.8 | 30.8 | 35.9 | 25.2 | 28.1 | 32.1 | 29.2 | 37.3 | 32.0 |
> >| ROTTA       | 49.1 | 45.0 | 45.5 | 30.2 | 42.6 | 29.5 | 26.0 | 32.3 | 30.6 | 37.6 | 24.7 | 29.2 | 32.7 | 30.4 | 36.9 | 34.8 |
> >| COTTA       | 40.6 | 38.0 | 39.9 | 27.2 | 38.2 | 28.5 | 26.5 | 33.3 | 32.2 | 40.6 | 25.1 | 26.9 | 32.3 | 28.3 | 33.8 | 32.8 |
> >| **AEA (ours)**  | 36.2 | 32.8 | 32.8 | 25.3 | 36.3 | 27.5 | 24.8 | 29.7 | 29.0 | 31.8 | 24.0 | 25.3 | 32.2 | 28.1 | 35.8 | **30.1** |
> >
> >&nbsp;
> >
> >Thanks again for your suggestion and we have included these additional results and details in the Appendix of our revised manuscript (SFDA in Sec. B.4 and continual setting in Sec. B.9).
> >
> >&nbsp;
> >
> >[1] Wang et al. Continual test-time domain adaptation. CVPR'22.

---

> > ### Author Response · Authors · 2024-11-22
> >
> > ### **Clarification on the addressed TTA setting**
> > >
> > >We apologize for the confusion. **The online TTA is one of the most common settings in TTA where the target batches are sequentially given to the model during test time.** In the paper, we consistently consider the online TTA and use the terminology 'TTA' interchangeably with online TTA (OTTA) throughout the paper (after line 142). For clarity, we have included the term ‘online’ in places and supplemented the related work section in our revised manuscript.
> >
> >
> > ### **Punctuation of equations. (3), (4), and (9)**
> > >
> > >Thank you for your comments. We have included punctuations to the equations mentioned in the revised manuscript.
> >
> > We are grateful to the reviewer for the helpful comments with a positive evaluation. If any concerns remain, we would welcome the opportunity to respond further.

---

> ### Author Response · Authors · 2024-12-01
>
> Dear Reviewer SzkH,
>
> We would like to express our gratitude once again for your constructive comments and positive evaluation of our work. As the discussion period is ending, we respectfully remind you of our response to further clarify the reviewer's concerns.
>
> In particular, following the reviewer's suggestions, we have (i) conducted additional experiments in other source-free scenarios, including the SFDA task (in Table 1) and the continual TTA setting (in Table 2), to further demonstrate the versatility of our AEA; (ii) clarified our TTA setup and expression of the equations in our paper. We have also included the raised points in our revised manuscript accordingly.
>
> If the reviewer has any remaining concerns or suggestions, we would be grateful for the opportunity to respond further. Thank you for your valuable feedback and suggestions.

---

### Official Review · Reviewer_V21s · 2024-11-04

**Soundness:** 3
**Presentation:** 4
**Contribution:** 3
**Rating:** 8
**Confidence:** 5

**Summary:**

This paper targeted at accelerating test-time adaptation from a new perspective, i.e., reducing the energy gap between the source and target domains. There are two key losses in the proposed method. The one is the source-free energy alignment (SFEA) loss to align the overall energy magnitude and the other is the logit cosine similarity (LCS) loss to guide the class-wise alignment direction. Experimental results on three common benckmarks (CIFAR10-C, CIFAR100-C, and TinyImageNet-C) have demonstrated the effectiveness of the proposed method.

**Strengths:**

The paper is easy-to-follow. The topic is essential yet the idea is moderate. Both test-time adaptation and energy-based models are important topics for the community and the paper addresses two formulations simultaneously.

**Weaknesses:**

- **Limited motivation.** My major concern is about the motivation of this work. First, the necessity of the proposed logit cosine similarity (LCS) loss is not clear. Second, though the authors have tried to describe the correlation between energy gap reduction and better adaptation, it's not clear why TTA needs to do this. Especially, many previous works have proven its effectivebess in addressing distribution shift problems .Third, the proposed method is complicated but the improvements in Tables 1,2,3 is minor.

- **Insufficient justifications.** From main results of Table 1,2,3, it can be observed that the proposed AEA obtains limited improvements in three small-scale image classification datasets (CIFAR10-C, CIFAR100-C, and TinyImageNet-C). Large-scale datasets and other tasks like segmentation or detection might be evaluated to demonstrate the AEA's utlization in various scenarios and applications.

**Questions:**

- How to define the directional aspects of energy alignment?

- It seems that the source-free energy alignment loss ($\mathcal{L}_{SFEA}$) is calculated in each target batch. How can this calculation method avoid the noise caused by bath sampling?

Minor comments:
- Actually, the foundamental of this work is to answer why does energy alignment lead to better adaptation. Thus, in my view, Sec. A.4 should be moved to the main context and the revisit of EMBs can be moved to the appendix.

**Details Of Ethics Concerns:**

None.

---

> ### Author Response · Authors · 2024-11-22
>
> ### **Motivation of our LCS loss**
> >
> >**Our LCS loss is necessary to address the directional aspect of energy alignment, particularly concerning the class-wise adjustment of logit elements.** Specifically, our SFEA loss alone minimizes the free energy of the target samples, aligning the energy between the source and target domains in terms of the magnitude of energy levels. That is, it increases the overall magnitude of the logits across all classes rather than considering the relativity of each logit element (i.e., class-wise correlation). On the other hand, **our LCS loss additionally accounts for class-wise correlation by increasing the logit elements of classes with high correlation together.** This allows each target sample to align with the well-trained source knowledge in a class-wise manner, leading to directionally improved energy alignment.
> >
> >&nbsp;
> >
> >To support our claim, in Fig. 9 of Appendix, we provide sample-wise visualizations (similar to Figs. 1-(a) and (b)) with and without our LCS loss. In the results, while our SFEA loss effectively reduces the energy gap, a number of target samples are still not clearly separated according to their true classes. In contrast, by incorporating our LCS loss, the target samples align more successfully with the source samples in a class-wise manner, guiding the direction of energy alignment in a logit space. We have supplemented these points in Sec. A.4 of the Appendix, and we appreciate your comments.
>
>
> ### **Energy gap reduction for better adaptation in TTA**
> >
> >We appreciate the reviewer for this comment. We note that TTA is one of the distribution shift problems where the source and target domain distributions are different. As we figured out in Fig. 1, this domain disparity causes the energy gap, especially pronounced in the early-stage of batch arrivals. Although there are many methods to address the problem of distribution shift, our energy gap reduction can **mitigate the domain disparity by promoting feature alignment between the source and target domains (as we discussed in Fig. 7).** Therefore, our adaptive energy alignment approach can lead to better and accelerated adaptation in TTA, compared to earlier methods that handle distribution shifts without considering energy gap reduction. We have supplemented this point in Sec. 4.3 and Sec. A.8 of our revised manuscript.
>
>
> ### **Regarding performance improvements**
> >
> >First, **we respectfully maintain that our AEA's performance improvement on average is meaningful.** In fact, there are a few notable papers that reported similar types of gains at the time of publication relative to prior works [1-2]. For example, regarding CIFAR10-C, [1] shows an improvement of approximately 0.8% over TENT, whereas our AEA achieves a gain of 4.5% over TENT. But more importantly, though, we would like to stress that the key advantage of our AEA is **its ability to accelerate adaptation**, as demonstrated in Fig. 5. That is, our proposed method is preferable in practical scenarios where the number of online batches is small, leading to competent early-stage performance in TTA. Also, since our AEA only modifies the loss function, it introduces merely negligible extra delay/resources, which provides a practical advantage.
> >
> >&nbsp;
> >
> >[1] Niu et al. Efficient test-time model adaptation without forgetting. ICML’22.
> >
> >[2] Goyal et al. Test time adaptation via conjugate pseudo-labels. NeurIPS’22.

---

> > ### Author Response · Authors · 2024-11-22
> >
> > ### **Justifications on large-scale datasets and other tasks**
> > >
> > >Thank you for your constructive suggestion. First, we provided the **results on large-scale datasets (i.e., ImageNet-C)** in Sec. B.1 of Appendix. To demonstrate the effectiveness of our AEA in terms of acceleration, we consider two different cases: (1) where there are sufficient target samples (50K, full), and (2) where the number of target samples is smaller (10K). The results show that our method achieves competitive performance in both cases. More importantly, in the case of limited target samples (10K), our AEA outperforms the baselines by significant margins (≥ 4.0%), supporting our claim that our AEA facilitates the acceleration of model adaptation in TTA.
> > >
> > >&nbsp;
> > >
> > >For further justification, as per the reviewer's suggestion, we also **evaluate our method on the segmentation TTA task** during the rebuttal period. In Table 1, we report the segmentation performance (i.e., MIoU(%, ↑)) on the Cityscapes dataset [1], following the experimental setup of [2]. Specifically, we adapt a pre-trained segmentation model (on GTA-5 datasets [3]) to target samples from Cityscapes datasets. The results show that our AEA still achieves competitive performance in the segmentation TTA task, demonstrating its effectiveness and applicability across various tasks. We have also supplemented the segmentation results (with experimental details) in Sec. B.3 of Appendix.
> > >
> > >&nbsp;
> > >- **Table 1. Performance (MIoU(%, ↑)) under the TTA segmentation task on Cityscapes dataset.**
> > >We consider five different target sequences from Cityscapes dataset following [2] and other experimental details are provided in Sec. B.3 of Appendix.
> > >
> > >| Target sequence         | Seq.#1 | Seq.#2 | Seq.#3 | Seq.#4 | Seq.#5 | Avg. |
> > >| ------------- | ------ | ---------- | ------ | --------------- | ---- | ---- |
> > >| No adaptation | 42.8   | 41.9       | 47.6   | 46.3            | 47.9 | 45.3 |
> > >| BN adaptation | 43.3   | 42.7       | 48.1   | 47.0            | 48.5 | 45.9 |
> > >| TENT          | 43.7   | 43.6       | 48.8   | 47.8            | 49.1 | 46.6 |
> > >| AEA (ours)    | **44.7**   | **44.9**       | **50.3**   | **48.3**            | **49.3** | **47.5** |
> > >
> > >&nbsp;
> > >
> > >[1] Cordts et al. The cityscapes dataset for semantic urban scene understanding. CVPR’16.
> > >
> > >[2] Volpi et al. On the road to online adaptation for semantic image segmentation. CVPR’22.
> > >
> > >[3] Richter et al. Playing for data: Ground truth from computer games. ECCV’16.
> >
> >
> > ### **Definition of the directional aspects of energy alignment**
> > >
> > >We note that **each target sample should be appropriately aligned in distinct directions based on its class information**, and we define this class-wise consideration as the **directional aspect of energy alignment.** For example, in Figure 1-(a) and (b), the energy gap for each target sample should be reduced in the direction corresponding to its class information (i.e., toward the same color). To achieve this, our LCS loss guides model adaptation to ensure that the class-wise correlation of the target domain aligns with that of the source domain during energy alignment.
> > >
> >
> >
> > ### **Avoidance to noise in SFEA loss**
> > >
> > >Thank you for your insightful comment. As the reviewer pointed out, our algorithm estimates the approximated source energy for each target batch to reduce the energy gap in a source-free scenario. Although randomness in batch sampling can include noisy samples within a batch, **our approximation is achieved by calculating the averaged energy over 'multiple' samples (i.e., the bottom α% of target samples within each batch), which dampens the samping noise.** This approach allows us to reduce target energy while being less affected by noisy samples. Furthermore, such noisy samples are likely to be more uncertain OOD instances, thus having relatively higher energy, which makes it less probable for them to be included in the sample set used for the approximation. To support our claim, in Fig.1-(c) and Fig.2, we measured the energy gap between the true source and target domains. The results show that our approach effectively reduces this gap, confirming the effectiveness of the proposed approximation strategy in the source-free scenario.
> >
> > ### **Rearranagement of the main context and appendix**
> > >
> > >Thank you for your constructive feedback. As the review's point, the key foundation of our work lies in the relationship between energy alignment and better adaptation. Following the reviewer’s suggestion, we have supplemented the relevant details to Sec. 4.3 of our revised manuscript.
> >
> > Again, thank you for your time and efforts in reviewing our paper. We would appreciate further opportunities to answer any remaining concerns you might have.

---

> ### Comment · Reviewer_V21s · 2024-11-26
> **Response to the rebuttal**
>
> Dear authors,
>
> I would appreciate your responses. I have read your rebuttal and I have the following additional remarks.
>
> >The impact of the idea.
>
> I pretty much agree with them that the proposed energy alignment is very similar to the earlier works[1,2]. And the LCS loss is no further insight for the readers. It is not ready for ICLR in its current version.
>
> >About additional results.
>
>  I really appreciate that so many experiments are added quickly.
>
> Ref: [1] Energy-based out-of-distribution detection. [2] Active learning for domain adaptation: An energy-based approach.

---

> > ### Author Response · Authors · 2024-11-26
> >
> > We appreciate the reviewer for follow-up comments and important remarks.
> >
> > 1. Regarding the impact of the idea, **our AEA has a distinct advantage in the source-free TTA scenario compared to the related papers [1, 2].** Specifically, [1] was proposed to amplify the energy gap (rather than decrease it) between ID and OOD samples for OOD detection, while [2] aimed to reduce the energy gap but required the entire source domain to be provided. Both works can only operate in scenarios where both the source and target domains (or ID and OOD) are provided in advance, making them unsuitable for TTA. In contrast, our approach is designed to **adaptively reduce the energy gap over time (strongly in the initial batches and more weakly later on) without requiring the source domain**, allowing it to address the challenging source-free TTA scenario. Moreover, compared to the hinge loss in [1, 2], our loss design (with the softplus function) has the additional advantages of being less sensitive to hyperparameters, along with a marginal performance gain, which further supports the compatibility of our method for the source-free TTA task. (We have discussed this point in Sec. A.6.)
> >
> > 2. In terms of LCS motivation, our key intuition is that **considering class-wise correlation during energy alignment allows the target features to be better aligned with the true source features in a class-wise manner.** In other words, our LCS loss encourages model adaptation so that logits for highly correlated classes are similar, while logits for low-correlated classes are dissimilar to each other. This turns out to **facilitate the feature alignment between the target and source domains in distinct directions based on their true class information**, as depicted in Fig. 9 of our revised manuscript. Therefore, we argue that our LCS loss is necessary to further enhance the energy alignment approach, leading to improved performance. If there is any aspect that still remains unclear, we would appreciate it if you could let us know. We will do our best to elaborate on the revewier's concerns and provide further insights regarding our LCS loss.
> >
> > Due to the above reasons, despite the existing works on energy gap reduction to solve the distribution shift problem, considering that (i) our work is the first to design a source-energy approximation method based on the target samples in a source-free TTA scenario, (ii) our work is the first to learn the target feature representations considering the class-wise correlation in a TTA setting, and (iii) our approach shows advantages even in the segmentation task (which is a task that has not been considered in most TTA research), we believe that our work can advance the field of test-time adaptation from the energy alignment perspective. We hope the reviewer can reevaluate our paper from these perspectives. Thank you once again for your comments.

---

> > > ### Comment · Reviewer_V21s · 2024-11-30
> > > **Decision for the rebuttal**
> > >
> > > Thank you for your reply. It would be better to discuss the similarities and dissimilarities with the related papers [1,2] in the next version. My concerns are addressed and I will raise my score towards acceptance.

---

> > > > ### Author Response · Authors · 2024-11-30
> > > >
> > > > We are truly delighted to have addressed the reviewer’s concerns. Thank you for your continuous feedback and valuable comments during the rebuttal period, which have improved our paper. Following the reviewer’s suggestion, we will better clarify the relationship between our work and prior studies [1, 2], including our unique contributions. Once again, thank you for recognizing our efforts.

---

### Official Review · Reviewer_aNYF · 2024-11-07

**Soundness:** 2
**Presentation:** 2
**Contribution:** 2
**Rating:** 3
**Confidence:** 4

**Summary:**

The paper introduces an energy-based approach for addressing the test-time adaptation (TTA) problem. The authors analyze the classical entropy minimization loss widely adopted by existing TTA methods through the lens of energy-based model, pinpointing the potential issue lies within the entropy minimization. To address the issue, two alignment objectives termed source-free energy alignment (SFEA) and logit cosine similarity (LCS) are proposed. SFEA aligns the overall energy magnitudes between source and target domains to reduce the energy gap during adaptation, whereas LCS ensures the class-wise correlation of the target domain maintains consistent during energy alignment. Experiments are conducted on several common TTA benchmarks including CIFAR10-C, CIFAR100-C, TinyImageNet-C and ImageNet-C, as well as style shift dataset PACS. The results of the proposed method generally surpass the baselines.

**Strengths:**

1.	As stated in the paper, Test-time adaptation is an important direction to mitigate the out-of-distribution issue of inference data and worths exploration.
2.	The experiment is extensive on several benchmarks, and the proposed method achieves good results on most tasks within these benchmarks.

**Weaknesses:**

1.	In both the summary and the introduction, the authors argue that “the EM loss can be decomposed into two energy-based terms with conflicting roles” and such confliction “hinders the energy gap reduction when the EM loss is used alone.” However, the authors do not address this issue in the proposed method. Minimizing the target energy does not resolve the conflict. On the contrary, the reviewer thinks that the free-energy maximization term in equation (4) is necessary for avoiding the explosion of the magnitude of logits. The reviewer’s opinion is supported by the fact that the final objective of the proposed method still incorporates EM loss.
2.	The authors highlight the strength of the proposed method that can reduce the energy gap more quickly than EM loss in the early batches. However, it appears from fig. 2 that the proposed method performs similarly to other baselines at the beginning. The result makes the author’s claim questionable.
3.	Given that previous works [1][2] adopt hinge loss for energy alignment, the reviewer suggests that the authors should conduct ablation study on the choice of loss to support their design.
4.	For each target batch, the proposed method computes a new approximated energy of the source domain, which means that the estimated source energy varies over time. This is a seemingly strange design without explanation. In fact, the reviewer suspects that this particular design is the actual reason for performance enhancement and distinguishes it from the previous energy alignment method. In reviewer’s opinion, the target energy is not actually aligned with the source energy, but rather is minimized towards an adaptively changing goal. This allows the energy minimization process to be more flexible and friendly to the model training.
5.	The second line of equation (4) is mistaken. It should include parentheses over the two energy terms.

Ref:
[1] Energy-based out-of-distribution detection.
[2] Active learning for domain adaptation: An energy-based approach.

**Questions:**

1.	In the caption of fig. 3, the ‘relative distance in a logit space’ is confusing. Please elaborate on the meaning of the x-axis in detail.
2.	From the experiment results, it appears that SHOT achieves better performance than most of the TTA methods. What is the reason?

---

> ### Author Response · Authors · 2024-11-22
>
> ### **Connection between the EM loss and our AEA**
> >$$
> >H(x; \theta) +L_{SFEA}(x, \theta)=
> >\sum_{j=1}^{K} p_\theta(y_j|x)E_{\theta}(x, y_j) - E_{\theta}(x) + \log(1+\exp(E_{\theta}(x)-\hat{E}_{s}))
> >$$
> >
> >Thank you for bringing up an important point. To provide a clear explanation, we present the summation of the EM loss and our SFEA loss above. First, in the EM loss, the first term encourages a reduction in energy, while the second term conversely encourages an increase in energy, resulting in conflicting roles for these two energy-based terms. Consequently, the EM loss alone has inherent limitations in sufficiently reducing the energy gap. To address these limitations, our SFEA loss (the third term) **explicitly minimizes target energy, partially offsetting the free-energy maximization term (the second term) of the EM loss.** This is why our method resolves the conflict and achieves better energy gap reduction compared to using the EM loss alone.
> >
> >&nbsp;
> >
> >Also, we agree with the reviewer’s point regarding the necessity of the free-energy maximization term, as it appropriately regularizes the adaptation after the energy gap has been sufficiently reduced. However, in the early-stage of batch arrivals where the energy gap between source and target domains is pronounced, this term may restrict sufficient reduction of the energy gap (as observed in Fig. 1-(c) of the main manuscript), which is the main motivation of introducing our SFEA loss. **Our SFEA loss is proposed to adaptively minimize target energy over time (strongly in the initial batches and more weakly later on), thereby reducing the energy gap more quickly.** In turn, our method is designed to complement the limitations of the EM loss while leveraging its advantages in TTA (rather than replacing it), which is why we incorporate EM loss in our final objective. We clarified these points in the revised manuscript (in line 251~253 and Sec. A.2).
>
> ### **Energy gap reduction in the early batches**
> >We appreciate the reviewer for this comment. In the results, at the beginning of the batch arrivals (i.e., ≤10 batches), the source pre-trained model has not been updated sufficiently to the target domain, which inevitably leads to minimal differences. Nevertheless, our AEA reduces the energy gap more than 6.4% compared to EM loss after only the initial 25 batches, which in turn leads to successful energy gap reduction in subsequent batches. Therefore, in TTA, it is important to **reduce the energy gap in the early batches**, and our adaptive energy alignment approach is effective in achieving this goal.
>
> ### **Ablation study on the choice of loss function**
> >Thank you for your constructive suggestion. In table 1, we conduct an ablation study on the choice of loss function, comparing the hinge loss to our SFEA loss. Specifically, we report the results with varying $\lambda_1$, a coefficient of $\mathcal{L}_{SFEA}$, to evaluate hyperparameter sensitivity. The results demonstrate that our loss is **less sensitive to hyperparameters than hinge loss, achieving better performance with marginal gains**. Our intuition is that these improvements are attributed to the smoothness of the softplus function around the near-zero region, containing more information useful for reducing the energy gap than the hinge loss near that region.
> >
> >&nbsp;
> >
> >We would also like to note that, unlike previous energy alignment works that have access to the source domain, the source-free scenario in TTA does not allow for accurate estimation of source energy. Under this new setting, we have utilized the approximated source energy for energy alignment, wherein **our smoothed loss design proves to be more feasible and leads to better performance compared to the hinge loss.** We have supplemented this ablation study with details in Sec. A.6.
> >
> >&nbsp;
> >
> >- **Table 1. Ablation study on the choice of loss function.**
> We report the average classification accuracy (%, ↑) on CIFAR10-C dataset. Experimental details are consistent with those in the paper.
> >
> >|   $\lambda_1$    | 1.0    | 1.2  | 1.4  | 1.6  | 1.8  | 2.0    |
> >| ----- | ---- | ---- | ---- | ---- | ---- | ---- |
> >| hinge | 80.3 | 80.2 | 79.9 | 79.7 | 79.3 | 79.0 |
> >| **ours**  | **80.5**(+0.2) | **80.7**(+0.5) | **80.6**(+0.7) | **80.2**(+0.5) | **79.9**(+0.6) | **79.8**(+0.8) |

---

> > ### Author Response · Authors · 2024-11-22
> >
> > ### **Regarding our approximated source energy approach in source-free scenario**
> > >
> > >Thank you for your insightful comment. As the reviewer pointed out, our algorithm minimizes the target energy towards an adaptively changing goal (i.e., approximated source energy). We note that, unlike previous energy alignment methods, our TTA setting **does not allow access to the source domain, making it infeasible to directly estimate source energy.** Although our scheme minimizes the target energy towards approximated source energy due to infeasibility of the access to source domain, **our method turns out to align the energy of source and target domains as the adaptation progresses.** To support our claim, in Fig.1-(c) and Fig.2, we measured the energy gap between the true source and target domains. The results show that our approach effectively reduces this gap, confirming the effectiveness of the proposed approximation strategy in the source-free scenario. We have clarified this point in the revised manuscript (in line 296~299).
> >
> > ### **Regarding the equation of EM loss decomposition**
> > >
> > >Thank you for your comments; however, there seems to be no error in equation (2) (it was originally equation (4) but changed to equation (2) in the revised version.). As the last term in the first line of that equation is not dependent on $j$, the sum of the outer probabilities is equal to 1. Therefore, that last term can be factored out, resulting in the energy term $E_\theta(x)$. To clarify, we provide a detailed derivation below and have explained the details in the revised manuscript (in line 216~218).
> > >
> > >$$
> > >\mathcal{H}(x; \theta) = \-\sum_{j=1}^{K} p_\theta(y_j|x) \Big(f_{\theta}(x)[y_j] - \log \sum_{i=1}^{K} e^{f_{\theta}(x)[y_i]}\Big) \\ = -\sum_{j=1}^{K} p(y_j|x) f_{\theta}(x)[y_j] + \log \sum_{i=1}^{K} e^{f_{\theta}(x)[y_i]} \\ = \sum_{j=1}^{K} p_\theta(y_j|x) E_{\theta}(x, y_j) - E_{\theta}(x)
> > $$
> >
> >
> > ### **Regarding the caption of Fig. 3**
> > >
> > >We apologize for the confusion we made. In Fig. 3, our intention in using 'relative distance' is to emphasize the changes in distance between the logit distributions to provide a high-level explanation of our method. To be more precise, the x-axis represents the **logit (projected) of each distribution**, and we have revised the caption of Fig. 3 and the x-axis in the manuscript accordingly.
> >
> > ### **Performance of SHOT in the results**
> > >
> > >Thank you for this comment. SHOT has the capability to learn a target-specific feature extractor through their source hypothesis transfer approach in a source-free scenario. Although SHOT was originally proposed to tackle a source-free domain adaptation task, it has demonstrated strong performance thanks to its feature extractor even in the online TTA setting within our fair evaluation benchmark [1] under consistent configurations (e.g., pre-trained model, learning rate, batch size, etc.) In such a fair evaluation setup, SHOT achieves competitive TTA performance, which is consistent with their results [1].
> > >
> > >&nbsp;
> > >
> > > [1] Zhao et al. On pitfalls of test-time adaptation. ICML’23.
> >
> >
> > We are truly grateful for the reviewer’s valuable and detailed feedback. We tried to carefully address the raised points in our responses, and have made clear the points in our revised manuscript. If there are any remaining concerns/suggestions, we would appreciate the opportunity to respond further.

---

> ### Author Response · Authors · 2024-12-01
>
> Dear Reviewer aNYF,
>
> Thank you for your valuable and detailed comments to improve our work. As the discussion period is coming to a close, we would like to kindly remind you of our response to address the reviewer’s concerns.
>
> Specifically, we have clarified (i) the relationship between the EM loss and our AEA, (ii) the energy gap reduction in the early batches, (iii) the choice of loss function, (iv) our approximated source energy approach in source-free TTA scenario and (v) other details regarding the equation of EM loss decomposition, the caption of Fig. 3, and the experimental results. We have also supplemented the raised points in our revised manuscript accordingly.
>
> We sincerely hope to provide further clarification and discuss any additional questions the reviewer may have. Once again, we greatly appreciate your time and effort.

---

### Author Response · Authors · 2024-11-25
**Global Comments**

We sincerely appreciate all reviewers for their insightful and constructive comments, which have greatly helped us improve our paper.
We believe that our responses have successfully addressed all concerns/questions raised by reviewers.
Below, we summarize the major comments by the reviewers along with our responses.

1. **Additional experiments on other tasks/settings**: We additionally evaluated our method on the segmentation TTA task (Reviewer V21s), source-free domain adaptation task (Reviewer SzkH), and continual TTA setup (Reviewer SzkH). We also supplemented each content in sections B.3, B.4 and B.9, respectively.
2. **Additional analysis and discussion**: We discussed the connection between the EM loss and our AEA (Reviewer aNYF) with clarification in line 251~253 and section A.2 of the revised manuscript. We also provided ablation study on the loss function (Reviewer aNYF, in section A.6) and analyzed how AEA achieves TTA acceleration with the additional results of feature alignment (Reviewer w8pg, in section A.8).
3. **Motivation of our AEA**: We elaborated the correlation between our SFEA loss and EM loss (Reviewer aNYF, in section A.2), and key motivation of our LCS loss (Reviewer V21s, in section A.4).
4. **Clarification**: We clarified (1) our approximated source energy approach in SFEA loss (Reviewer aNYF and V21s), (2) how our energy gap reduction leads to better adaptation/acceleration in TTA (Reviewer aNYF and V21s). We also made these points clear in the line 296~299 for (1) and in sections 4.3 and A.8 for (2) in our revised manuscript.

If there are any remaining issues, we would appreciate further chances to respond. We would also be grateful if you would consider reevaluating your scores.

Sincerely, \
Authors

---

### Meta-Review · Area_Chair_mNuH · 2024-12-22

**Metareview:**

This paper proposes the novel SFEA and LCS losses, forming an effective AEA solution to accelerate TTA. The paper is well-written and easy to follow. The authors conducted thorough comparative experiments to evaluate the effectiveness of the proposed method, along with an ablation study to assess the impact of its components. Additionally, they provided the code and datasets in the supplementary material. During the rebuttal phase, the authors addressed most of the reviewers' concerns, leading two reviewers to increase their ratings to acceptance. The paper received final scores of 3, 8, 6, and 8. Based on the overall reviewers' feedback, this paper is recommended for acceptance. However, the authors are encouraged to further address the remaining issues in the final version, such as discussing the similarities and differences with related works [1,2].

**Additional Comments On Reviewer Discussion:**

Three reviewers participated in the discussions with the authors and their concerns have been addressed.

---

### Decision · Program_Chairs · 2025-01-22

Accept (Poster)